# START: A Generalized State Space Model with Saliency-Driven Token-Aware Transformation

**Jintao Guo**[1]  **Lei Qi**[2*]  **Yinghuan Shi**[1*]  **Yang Gao**[1†]

[1] Nanjing University   [2] Southeast University

guojintao@smail.nju.edu.cn, qilei@seu.edu.cn, {syh, gaoy}@nju.edu.cn

## Abstract

Domain Generalization (DG) aims to enable models to generalize to unseen target domains by learning from multiple source domains. Existing DG methods primarily rely on convolutional neural networks (CNNs), which inherently learn texture biases due to their limited receptive fields, making them prone to overfitting source domains. While some works have introduced transformer-based methods (ViTs) for DG to leverage the global receptive field, these methods incur high computational costs due to the quadratic complexity of self-attention. Recently, advanced state space models (SSMs), represented by Mamba, have shown promising results in supervised learning tasks by achieving linear complexity in sequence length during training and fast RNN-like computation during inference. Inspired by this, we investigate the generalization ability of the Mamba model under domain shifts and find that input-dependent matrices within SSMs could accumulate and amplify domain-specific features, thus hindering model generalization. To address this issue, we propose a novel SSM-based architecture with saliency-based token-aware transformation (namely START), which achieves state-of-the-art (SOTA) performances and offers a competitive alternative to CNNs and ViTs. Our START can selectively perturb and suppress domain-specific features in salient tokens within the input-dependent matrices of SSMs, thus effectively reducing the discrepancy between different domains. Extensive experiments on five benchmarks demonstrate that START outperforms existing SOTA DG methods with efficient linear complexity. Our code is available at https://github.com/lingeringlight/START.

## 1 Introduction

Deep learning models have achieved impressive progress in various computer vision tasks over the past years [1–3]. Such a huge success is mostly based on the *independent and identically distributed* (*i.i.d.*) assumption, *i.e.*, the training and testing data follow the same distribution [4]. However, when evaluated on test data following different distributions from the training data, these models often suffer severe performance degradation. This issue, which is known as domain shift [4], has greatly hindered the applications of deep learning models in the real world.

To improve the generalization of the model under domain shifts, Domain Adaptation (DA) has been widely studied, which aims to transfer the knowledge learned from labeled source domains to the unlabeled or partially labeled target domain [5, 6]. However, DA methods cannot guarantee the performance of the model on unknown target domains that have not been observed during training [7, 8]. Since the accessibility of the target domain could not always be satisfied in real scenarios,

---

[*]Corresponding authors: Yinghuan Shi and Lei Qi.

[†]Jintao Guo, Yinghuan Shi, and Yang Gao are with the National Key Laboratory for Novel Software Technology and the National Institute of Healthcare Data Science, Nanjing University.

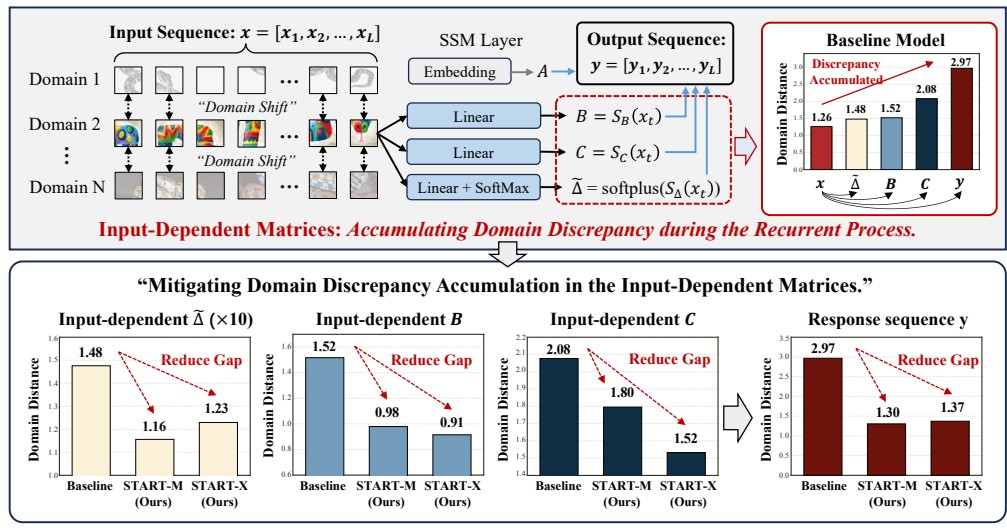

Figure 1: **Analysis of the input-dependent matrices in SSMs.** We investigate domain discrepancy in the input sequence $x$, response sequence $y$, and the input-dependent matrics $\tilde{\Delta}$, $B$, and $C$. The results indicate that *the input-dependent matrices can accumulate the domain-specific features during the recurrent process, potentially increasing domain gap.* We experiment on PACS [24] with Sketch as the target domain, analyzing the representations from the last block of VMamba backbone [22].

Domain Generalization (DG) is proposed to develop a domain-generalizable model on unseen target domains by learning multiple different but related source domains [8, 9].

Most existing DG methods focus on learning domain-invariant representations across source domains, primarily via domain alignment [10, 11], meta-learning [7, 12], and data augmentation [13, 14]. These methods heavily rely on convolutional neural networks (CNNs), which have limited receptive fields due to local convolutions. Consequently, the CNN-based methods inevitably tend to learn local texture information, leading to overfitting to source domains and poor generalization on target domains[15, 16]. Recent works in DG have introduced Vision Transformers (ViTs) as the backbone for DG, utilizing the global receptive field of the self-attention mechanism to mitigate local texture bias [17–19]. However, the complexity of self-attention increases quadratically with input length, resulting in significant computational overhead for ViTs when modeling long sequences [20, 21].

To address this issue, some pioneers have proposed advanced state space models (SSMs) [20, 22, 23], represented by Mamba [20], which selectively models token dependencies in input sequences in a compressed state space. The selective scan mechanism allows Mamba to achieve linear complexity in sequence length during training and fast RNN-like computation during inference. Despite the remarkable performance of Mamba-based methods on supervised learning tasks, few existing works have analyzed the generalization ability of Mamba under domain shift. It remains an open question whether the Mamba model can achieve excellent performance for DG tasks.

In this paper, we theoretically analyze the generalization error bound of the Mamba model under domain shifts. We find that the domain distance of features extracted by the model is strongly related to the input-dependent matrices within the model. *These matrices accumulate and amplify domain-specific features during training, which exacerbates the overfitting issue of the model to source domains.* We empirically measure the distance among source domains within the input sequence $x$, the response sequence $y$, and the input-dependent matrices of the last network layer. As shown in Fig. 1, we observe that for the baseline model, input-dependent matrices ($\tilde{\Delta}$, $B$, and $C$) are prone to learning domain-specific features from the input $x$. Since the output $y$ is calculated by the recurrent product of $x$ and these matrices, domain-specific features are accumulated and amplified, causing the model to overfit the source domains. To address this issue, we propose a Generalized State Space Model with Saliency-driven Token-Aware Transformation (START), which can reduce domain-specific information in input-dependent matrices during training. Building on the latest Mamba-based model [22], we develop a strong baseline for DG that outperforms many SOTA DG methods, which selectively learns global dependencies among tokens with linear complexity in sequence length.

Moreover, based on theoretical analysis, we design the saliency-driven token-aware transformation method, which simulates domain shifts during training by selectively introducing style perturbations to tokens focused on by the input-dependent matrices. START constrain these matrices to learn domain-invariant features, thus mitigating the overfitting of model on source domains. Experiments on five datasets prove the effectiveness of our method. Our contributions are summarized as follows:

- We conduct a theoretical investigation into the generalization ability of the Mamba model, revealing that the input-dependent matrices in Mamba can accumulate domain-specific features during the recurrent process, thus hindering the model's generalizability.

- Based on theoretical analysis, we propose a novel SSM-based architecture with saliency-driven token-aware transformation as a competitive alternative to CNNs and ViTs for DG, which performs excellent generalization ability with efficient linear complexity.

- For the saliency-driven token-aware transformation, we explore two variants to identify and perturb salient tokens in feature sequences, effectively reducing domain-specific information within the input-dependent matrices of Mamba. Our method achieves SOTA performances, *e.g.*, yielding the best CNN-based method by $5.87\%$ ($58.27\%$ vs. $52.40\%$) on TerraIncognita.

## 2 Related Works

**Domain generalization.** Traditional DG methods, primarily based on CNN backbones, can be broadly classified into three categories: domain alignment, meta-learning, and data augmentation. Motivated by the learning theory of domain adaptation [4, 25], domain alignment methods seek to learn domain-invariant representations through adversarial learning [10, 26, 27], causal learning [28, 29], or feature disentanglement [30, 31]. Another popular way to address DG is meta-learning, which partitions the training data from multiple source domains into meta-train and meta-test sets to simulate domain shifts during training [7, 12, 32]. Data augmentation is also an effective method to enhance model robustness to domain shifts by generating diverse data invariants through adversarial generation [33, 34], style perturbation [13, 14], and learnable parameters [35, 36]. However, CNN-based DG methods suffer from the limited receptive field of convolutions, often leading to a texture bias and overfitting to source domains [15, 37]. To address this, some researchers have introduced ViT-based methods for DG, which capture global representations by leveraging long-range spatial dependencies with attention mechanisms [18, 17, 19]. Despite their advantages, ViT-based methods are computationally intensive due to the quadratic complexity of the self-attention mechanism, limiting their practical applications [20, 21]. Inspired by the emerging Mamba model [20, 22, 23], we explore a novel SSM-based architecture for DG that combines strong generalizability with efficient linear complexity. We theoretically analyze the generalization error bound of Mamba and design a novel saliency-driven token-aware transformation to suppress domain-specific features in the input-dependent matrices of Mamba, thereby enhancing the generalization ability of the model.

**State space models.** Recently, state space models (SSMs) have demonstrated promising performance across various vision tasks [38–40] for their ability to effectively capture long-range dependencies while maintaining linear scalability with sequence length. Derived from the classical state space model [41], the Structured State Space Sequence Model (S4) [42] addresses computational constraints through novel parameterizations catering to continuous-time, recurrent, and convolutional views of the state space model. Notably, Mamba [20] has emerged as a standout performer, which integrates selection mechanism and hardware-aware algorithms into previous works [43–45], thus achieving linear-time inference and efficient training mechanisms. Based on the success of Mamba, Vision Mamba (Vim) [23] applies Mamba to ViT architecture, combining bidirectional SSM for data-dependent global visual context modeling. Meanwhile, VMamba [22] designs a cross-scan module to bridge the gap between 1D array scanning and 2D plain traversing. Mamba-based architectures have exhibited superior performance across various supervised vision tasks, including medical image segmentation [46–48], point cloud analysis [49–51], and remote sensing analysis [52, 53]. However, few works explore the performance of Mamba under domain shifts for DG. Although DGMamba [54] has recently introduced a pioneering Mamba-based framework for DG, it lacks a deep analysis of the generalizability of Mamba. In the paper, we conduct a theoretical analysis of Mamba's generalizability, revealing that input-dependent matrices within Mamba could accumulate domain-specific information, thereby impeding model generalization. Consequently, we propose a generalized SSM-based architecture for DG, incorporating a non-parametric module to selectively perturb salient tokens within input-dependent matrices, thus enhancing model generalization to unseen domains.

# 3 Method

## 3.1 Preliminary

**State Space Models (SSMs).** The SSM is a type of linear time-invariant systems that map input sequence $x_t \in \mathbb{R}^L$ to response sequence $y_t \in \mathbb{R}^L$ through a hidden state $h_t \in \mathbb{R}^N$. Mathematically, this process is formulated as the subsequent linear ordinary differential equations (ODEs): $h_t' = Ah_t + Bx_t, y_t = Ch_t$, where $A \in \mathbb{R}^{N \times N}$ is the evolution parameter, and $B \in \mathbb{R}^{N \times 1}$, $C \in \mathbb{R}^{N \times 1}$ are the projection parameters. However, the differential equation is hard to solve in the deep learning setting, thus discrete SSM [55, 44] suggests discretizing the system with a time scale parameter $\Delta$:

$$
\begin{aligned}
\bar{A} &= e^{\Delta A}, \quad \bar{B} = (\Delta A)^{-1}(e^{\Delta A} - I) \cdot \Delta B, \\
h_t &= \bar{A}h_{t-1} + \bar{B}x_t, \quad y_t = Ch_t,
\end{aligned}
\tag{1}
$$

where $\bar{A}$ and $\bar{B}$ are discrete counterparts of the continuous parameters $A$ and $B$, and $\Delta \in \mathbb{R} > 0$ is the sampling timescale for the discretization process. Although the discrete SSMs can achieve linear time complexity, they rely on static parameterization, *i.e.*, $\bar{A}$, $\bar{B}$, and $C$ are time-invariant for any input, inherently limiting their ability to capture sequence context [20]. To address this issue, recently, [20] proposes Mamba, a selective SSM (S6) that effectively selects relevant context by enabling dependence of the parameters $B \in \mathbb{R}^{L \times N}$, $C \in \mathbb{R}^{L \times N}$, and $\Delta \in \mathbb{R}^{L \times D}$ on the input $x_t \in \mathbb{R}^{L \times D}$:

$$
B = S_B(x_t), \quad C = S_C(x_t), \quad \Delta = \text{softplus}(S_\Delta(x_t)).
\tag{2}
$$

$S_B$, $S_C$, $S_\Delta$ are linear projection layers and $\text{softplus}(\cdot) = \log(1 + \exp(\cdot))$. The input-dependent time-variant layers could enhance recurrent layers, making them more expressive and flexible in capturing complex dependencies [21]. The parameter matrixes can be further expressed as $\bar{A} = [\bar{A}_1, \cdots, \bar{A}_L]$, $\bar{B} = [\bar{B}_1, \cdots, \bar{B}_L]$, $C = [C_1, \cdots, C_L]$, where $L$ is the sequence length. Considering the initial state $h_0 = 0$, Eq. (1) can be unrolled as [21]:

$$
y = \alpha x, \quad
\begin{bmatrix} y_1 \\ y_2 \\ \vdots \\ y_L \end{bmatrix} =
\begin{bmatrix}
C_1\bar{B}_1 & 0 & \cdots & 0 \\
C_2\bar{A}_2\bar{B}_2 & C_2\bar{B}_2 & \cdots & 0 \\
\vdots & \vdots & \ddots & 0 \\
C_L\prod_{k=2}^{L}\bar{A}_k\bar{B}_L & C_L\prod_{k=3}^{L}\bar{A}_k\bar{B}_L & \cdots & C_L\bar{B}_L
\end{bmatrix}
\begin{bmatrix} x_1 \\ x_2 \\ \vdots \\ x_L \end{bmatrix},
\tag{3}
$$

where $\alpha_{i,j} = C_i \prod_{k=j+1}^{i} \bar{A}_k \bar{B}_j$, for $0 \le j < i \le L$, characterizing S6 layer as a data-dependent self-attention [56]. The attention matrix $\alpha$ is determined by both the input and the parameter matrices.

## 3.2 Theoretically Analysis for the Generalization Ability of Mamba

Previous DG methods have primarily focused on enhancing the generalizability of CNNs or ViTs, lacking theoretical investigations into the Mamba model. We theoretically explore the generalization error bound of Mamba, proving that *perturbing the domain-specific features within the input-dependent matrices of Mamba can effectively diminish the upper bound of the model's generalization risk* .

**Notations.** Given a training set of $N$ source domains $\mathcal{D}_S = \{D_S^1, D_S^2, \cdots, D_S^N\}$, the objective of DG is to use $\mathcal{D}_S$ to train a model that is expected to perform well on unseen target domain $D_T$. Let $h : \mathcal{X} \to \mathcal{Y}$ be a hypothesis from the candidate hypothesis space $\mathcal{H}$, where $\mathcal{X}$ and $\mathcal{Y}$ denote the input space and the label space, respectively. Since Mamba learns dependencies among tokens from continuous sequences, we study its generalizability at the token level. Let $\psi(\cdot)$ be the feature extractor of $h$ that maps input images into feature space. Following Integral Probability Metrics [57, 58], we define the token-level Maximum Mean Discrepancy to estimate the gap between different domains.

**Definition 1 (Token-level Maximum Mean Discrepancy).** *Given two different distributions of $D_S$ and $D_T$, let $L$ denote the number of tokens in the response sequence of $\psi(\cdot)$, then we define the TOken-level Maximum Mean Discrepancy (To-MMD) between $\psi(D_S)$ and $\psi(D_T)$ as:*

$$
d_{\text{To-MMD}}(D_S, D_T) = \frac{1}{L} \sum_{t=1}^{L} \sup_{\psi_t \in \Psi_t} \sup_{||f||_{\mathcal{F}_k} \le 1} \left| \int f d(\psi_t(D_S) - \psi_t(D_T)) \right|,
\tag{4}
$$

*where $\Psi$ represents the hypothesis space for each token, $\psi_t(D)$ denotes the distribution of the $t$-th token for domain $D$, and $\mathcal{F}_k$ is a RKHS with its associated kernel $k$.*

We here investigate the generalization risk bound of the Mamba model. Theoretically, as in [59, 60], the risk of the hypothesis $h$ on the domain $D$ is defined as: $R_D(h) = \mathbb{E}_{x \sim D}[\mathcal{L}(h(x) - h^*(x))]$, where $\mathcal{L} : \mathcal{Y} \times \mathcal{Y} \rightarrow \mathcal{R}_+$ is a convex loss-function that measures the distance between $h$ and the true labeling function $h^*$. Moreover, following [8, 60], for multiple source domains $\mathcal{D}_S = \{D_S^1, D_S^2, ..., D_S^N\}$, the convex hull $\Lambda_S$ is defined as a set of a mixture of source domains, i.e., $\Lambda_S = \{\bar{\mathcal{D}} : \bar{\mathcal{D}}(\cdot) = \sum_{n=1}^{N} \pi_i D_s^n(\cdot), \sum_{n=1}^{N} \pi_n = 1, \pi_n \in [0, 1]\}$. The $\bar{D}_T \in \Lambda_S$ is defined as the closest domain to the target domain $D_T$. Based on Eq. (4), the following generalization risk bound can be derived.

**Theorem 1 (Generalization Risk Bound).** *With the previous setting and assumptions, let $D_S^i$ and $D_T$ be two sets with $M$ samples independently drawn from $\mathcal{D}_S^n$ and $\mathcal{D}_T$, respectively. For any $\delta \in (0, 1)$ with probablity of at least $1 - \delta$, for all $h \in \mathcal{H}$, the following inequality holds:*

$$R_{D_T}(h) \leq \sum_{n=1}^{N} \pi_n R_{D_S}^n(h) + d_{\text{To-MMD}}(D_T, \bar{D}_T) + \sup_{i,j \in [N]} d_{\text{To-MMD}}(D_S^i, D_S^j) + 2\lambda_\pi + \sigma, \quad (5)$$

*where $\lambda_\pi = \frac{1}{M}(\sum_{n=1}^{N} \pi_n \mathbb{E}_{x \sim D_S^n}[\sqrt{tr(K_{D_S^n})}] + \mathbb{E}_{x \sim D_T}[\sqrt{tr(K_{D_T})}]) + \sqrt{\frac{log(2/\epsilon)}{2M}}$, and $\sigma$ is the minimum combined error of the ideal hypothesis $h^*$ on both $D_S$ and $D_T$. Let $\kappa_T = d_{\text{To-MMD}}(D_T, \bar{D}_T)$ and $\kappa_S = \sup_{i,j \in [N]} d_{\text{To-MMD}}(D_S^i, D_S^j)$, respectively.*

The proof of Theorem 1 is provided in Appendix A.1. The inequality indicates that the generalization error bound depends on $\kappa_T$ denoting the token-level maximum distance between source and target domains, and $\kappa_S$ measuring the maximum pairwise gap among source domains at the token level. The smaller the two terms, the lower the upper bound of generalization error. Following [25, 58, 61], we simplify the To-MMD in Eq. (4) by choosing a unit ball in the $\mathcal{F}_k$ and using Gaussian kernel with parameter $\gamma$ to estimate $\kappa_T$ and $\kappa_S$. Let $\bar{x}^S \in \mathbb{R}^L$ and $\bar{x}^T \in \mathbb{R}^L$ to denote the mean embeddings of samples from $D_S$ and $D_T$, where $L$ represents the token sequence length. We explore a simplified problem in conjunction with a single S6 layer, i.e., $\bar{y} = \alpha\bar{x}$, with $\alpha \in \mathbb{R}^{L \times L}$ being the data-dependent matric. The domain distance between $\bar{y}^S$ and $\bar{y}^T$ is formulated as $k(\bar{y}^S, \bar{y}^T) = \exp(-||\bar{y}^S - \bar{y}^T||^2/\gamma)$, where $\gamma$ is the kernel parameter. Specifically, for input-dependent matrices $B, C, \Delta$, we denote $\text{softmax}(S_\Delta(\cdot))$ as $\tilde{S}_\Delta(\cdot)$. Then, we analyze the impact of these input-dependent matrices on $||\bar{y}^S - \bar{y}^T||^2$, which is applicable to both $\kappa_T$ and $\kappa_S$. For the $i$-th tokens $\bar{x}_i^S$ and $\bar{x}_i^T$, we define:

$$\begin{aligned} d_{C\tilde{\Delta}Bx}(\bar{x}_i^S, \bar{x}_i^T) &= S_C(\bar{x}_i^S)\tilde{S}_\Delta(\bar{x}_i^S)S_B(\bar{x}_i^S)\bar{x}_i^S - S_C(\bar{x}_i^T)\tilde{S}_\Delta(\bar{x}_i^T)S_B(\bar{x}_i^T)\bar{x}_i^T, \\ d_{\tilde{\Delta}}(\bar{x}_i^S, \bar{x}_i^T) &= \tilde{S}_\Delta(\bar{x}_i^S) - \tilde{S}_\Delta(\bar{x}_i^T). \end{aligned} \quad (6)$$

With the recurrent property of the S6 layer in Eq. (3), we can derive the following propositions:

**Proposion 1 (Accumulation of Domain Discrepancy).** *Given two distinct domains $D_S$ and $D_T$, the token-level domain distance $d_{\text{To-MMD}}(D_S, D_T)$ depends on $d_{C\tilde{\Delta}Bx}(\bar{x}_i^S, \bar{x}_i^T)$ and $d_{\tilde{\Delta}}(\bar{x}_i^S, \bar{x}_i^T)$ for the $i$-th token. For the entire recurrent process, domain-specific information encoded in $S_\Delta$, $S_C$, and $S_B$ will accumulate, thereby amplifying domain discrepancy.*

**Proposion 2 (Mitigating Domain Discrepancy Accumulation).** *Perturbing domain-specific features in tokens focused on by $S_\Delta$, $S_C$, and $S_B$ can enhance their learning of domain-invariant features, thus effectively mitigating the accumulation issue in these input-dependent matrices.*

Propositions 1 and 2 are proved in Appendix A.1. Based on the propositions, we develop a saliency-driven token augmentation method, which perturbs style information within the tokens that the model focuses on at the sequence level. In this way, our method enhances the extraction of domain-invariant features by the input-dependent matrices, i.e., $S_\Delta$, $S_C$, and $S_B$. As presented in Tab. 6, we also empirically validate the effectiveness of our method in reducing domain discrepancy in these matrices.

### 3.3  Saliency-driven Token-Aware Transformation for Mamba

To boost the generalization ability of the Mamba model, leveraging the **Proposion 1** and **Proposion 2**, we propose a novel *Saliency-driven Token-AwaRe Transformation paradigm* (START in short), which aims to explicitly suppress domain-related features within the input-dependent matrixes. Unlike prior methods that perturb entire feature maps at the channel level [13–15], START incorporates a

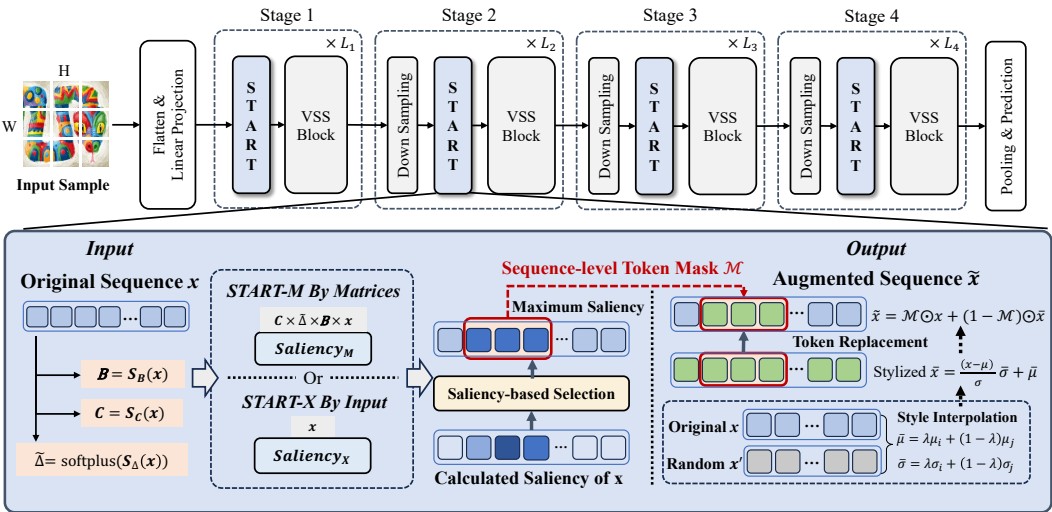

Figure 2: **Overall Architecture of the Proposed START Framework**. The core of the START framework is the Saliency-driven Token-Aware Transformation, which uses a saliency-driven scheme to localize tokens targeted by input-dependent matrices, subsequently perturbing domain-specific style information within these tokens. We designed two variants: START-M, which uses input-dependent matrices, and START-X, which uses input sequences to compute saliency.

saliency-driven token selection scheme to perturb the prominent regions of input-dependent matrics $S_\Delta$, $S_B$, and $S_C$. Based on the attention mechanism outlined in Eq. (3), we propose two variants to identify and perturb tokens within salient regions, including START-M that determines saliency using input-dependent matrices, and START-X computing saliency based on input sequences.

**START based on input-dependent matrices (START-M).** As **Proposition 1** reveals, for the $i$-th token, the token-level domain gap depends on $d_{C\tilde{\Delta}Bx}(\bar{x}_i^S, \bar{x}_i^T)$ and $d_{\tilde{\Delta}}(\bar{x}_i^S, \bar{x}_i^T)$. Specifically, as presented in Eq. (6), $d_{C\tilde{\Delta}Bx}$ is contingent on $S_C(x_i)\tilde{S}_\Delta(x_i)S_B(x_i)x_i$, which is the response of the SSM to $x_i$. $S_C(x_i)\tilde{S}_\Delta(x_i)S_B(x_i)$ could be regarded as a self-attention matrix, which implicitly offers a measure of saliency for a token $x_i$. To this end, we propose START-M, which utilizes the input-dependent matrices to identify salient tokens. Concretely, given an input sequence $\{x_i\}_{i=1}^L$, where $L$ denotes the sequence length, we first compute the input-dependent matrices based on Eq. (2). Then, we calculate the saliency value for each token based on Eq. (6), *i.e.*, for the $i$-th token $x_i$:

$$Saliency_M(x_i) = S_C(x_i)\text{softmax}(S_\Delta(x_i))S_B(x_i)x_i, \tag{7}$$

Afterward, we generate a binary mask $\mathcal{M}_S \in \mathbb{B}^L$ with the element being set to 1 if the corresponding element $Saliency_M(x_i)$ is in the top $P_{tokens}$ percentage elements.

Meanwhile, we synthesize the style-augmented sequence, which is achieved by mixing the mean and variance of different samples. Following [13, 62], we first compute the style statistics as: $\mu(x) = \frac{1}{L}\sum_{i=1}^L x_i, \sigma(x) = \sqrt{\frac{1}{L}\sum_{i=1}^L (x_i - \mu(x))^2}$. Then we randomly select another sample $x'$ from the current batch, utilizing its statistics to synthesize the stylized version of $x$:

$$\tilde{\mu} = \epsilon\mu(x) + (1-\epsilon)\mu(x'), \quad \tilde{\sigma} = \epsilon\sigma(x) + (1-\epsilon)\sigma(x'),$$

$$\epsilon \sim Beta(0.1, 0.1), \quad \tilde{x} = \frac{x - \mu(x)}{\sigma(x)} \cdot \tilde{\mu} + \tilde{\sigma}, \tag{8}$$

Finally, with the token-level mask $\mathcal{M}_S$, we mix $x$ and $\tilde{x}$ to generate the augmented sequence $x_{\text{aug}}$, where tokens with maximum saliency are style-augmented, while other tokens remain unchanged:

$$x_{\text{aug}} = \mathcal{M}_S \odot x + (1 - \mathcal{M}_S) \odot \tilde{x}, \tag{9}$$

where $\odot$ is element-wise multiplication. Note that when $P_{token} = 1$, START-M degenerates into a channel-level augmentation, *i.e.*, MixStyle [13]. However, note that style statistics could be one kind

Table 1: Performance (%) comparisons with the SOTA DG methods on PACS and OfficeHome.

| | | PACS | | | | | Office-Home | | | | |
|---|---|---|---|---|---|---|---|---|---|---|---|
| Method | Params. | Art | Cartoon | Photo | Sketch | Avg. | Art | Clipart | Product | Real | Avg. |
| CNN: ResNet-50 | | | | | | | | | | | |
| DeepAll [65] (AAAI'20) | 23M | 84.70 | 80.80 | 97.20 | 79.30 | 85.50 | 61.30 | 52.40 | 75.80 | 76.60 | 66.50 |
| PCL [66] (CVPR'22) | 23M | 90.20 | 83.90 | 98.10 | 82.60 | 88.70 | 67.30 | 59.90 | 78.70 | 80.70 | 71.60 |
| EoA [67] (NeurIPS'22) | 23M | 90.50 | 83.40 | 98.00 | 82.50 | 88.60 | 69.10 | 59.80 | 79.50 | 81.50 | 72.50 |
| EQRM [68] (NeurIPS'22) | 23M | 86.50 | 82.10 | 96.60 | 80.80 | 86.50 | 60.50 | 56.00 | 76.10 | 77.40 | 67.50 |
| SAGM [69] (CVPR'23) | 23M | 87.40 | 80.20 | 98.00 | 80.80 | 86.60 | 65.40 | 57.00 | 78.00 | 80.00 | 70.10 |
| iDAG [70] (ICCV'23) | 23M | 90.80 | 83.70 | 98.00 | 82.70 | 88.80 | 68.20 | 57.90 | 79.70 | 81.40 | 71.80 |
| DomainDrop [60] (ICCV'23) | 23M | 89.82 | 84.22 | 98.02 | 85.98 | 89.51 | 67.33 | 60.39 | 79.05 | 80.22 | 71.75 |
| CCFP [71] (ICCV'23) | 23M | 87.50 | 81.30 | 96.40 | 81.40 | 86.60 | 63.70 | 55.50 | 77.20 | 79.20 | 68.90 |
| MADG [72] (NeurIPS'23) | 23M | 87.80 | 82.20 | 97.70 | 78.30 | 86.50 | 67.60 | 54.10 | 78.40 | 80.30 | 70.10 |
| PGrad [73] (ICLR'23) | 23M | 87.60 | 79.10 | 97.40 | 76.30 | 85.10 | 64.70 | 56.00 | 77.40 | 78.90 | 69.30 |
| AGFA [74] (ICLR'23) | 23M | 89.80 | 85.20 | 97.60 | 84.70 | 89.30 | 67.50 | 58.50 | 79.30 | 80.70 | 71.50 |
| GMDG [75] (CVPR'24) | 23M | 84.70 | 81.70 | 97.50 | 80.50 | 85.60 | 68.90 | 56.20 | 79.90 | 82.00 | 70.70 |
| ViT-based or MLP-like models | | | | | | | | | | | |
| MLP-B [76] (NeurIPS'21) | 59M | 85.00 | 77.86 | 94.43 | 65.72 | 80.75 | 63.45 | 56.31 | 77.81 | 79.76 | 69.33 |
| SDViT [18] (ACCV'22) | 22M | 87.60 | 82.40 | 98.00 | 77.20 | 86.30 | 68.30 | 56.30 | 79.50 | 81.80 | 71.50 |
| ResMLP-S [77] (TPAMI'22) | 40M | 85.50 | 78.63 | 97.07 | 72.64 | 83.46 | 62.42 | 51.94 | 75.40 | 77.21 | 66.74 |
| ViP-S [78] (TPAMI'22) | 25M | 88.09 | 84.22 | 98.38 | 82.41 | 88.27 | 69.55 | 61.51 | 79.34 | 83.11 | 73.38 |
| GMoE-S [19] (ICLR'23) | 34M | 89.40 | 83.90 | 99.10 | 74.50 | 86.70 | 69.30 | 58.00 | 79.80 | 82.60 | 72.40 |
| SSM-based models | | | | | | | | | | | |
| DGMamba [54] (ACM MM'24) | 22M | 91.30 | 87.00 | 99.00 | 87.30 | 91.20 | **76.20** | 61.80 | 83.90 | 86.10 | 77.00 |
| Strong Baseline [22] | 22M | 91.55 | 85.11 | 99.14 | 83.97 | 89.94±0.52 | 75.06 | 60.48 | 84.71 | 85.45 | 76.43±0.15 |
| START-M (Ours) | 22M | **93.29** | **87.56** | 99.14 | 87.07 | **91.77±0.40** | 75.15 | 62.04 | **85.31** | **85.84** | **77.09±0.16** |
| START-X (Ours) | 22M | 92.76 | 87.43 | **99.22** | **87.46** | 91.72±0.49 | 75.48 | **62.06** | 85.24 | 85.47 | 77.07±0.07 |

of domain-specific feature, while other forms of domain-specific features may also exist, especially within the image backgrounds [59, 63]. As a result, directly perturbing the style information of tokens on the background might activate other forms of domain-related noise, which could still disrupt the model generalization [64]. To address this issue, our START-M proposes to selectively perturb tokens with the highest saliency, which are typically associated with foregrounds, thus enhancing the model learning of domain-invariant information without activating domain-related noise. Ablation study in Section 4.3 also proves the effectiveness of the saliency-driven selection scheme.

**START based on input sequences (START-X).** Based on **Proposition 2**, recalling that $\Delta$, $B$, and $C$ are all input-dependent matrices (as in Eq. (2)), we design a simplified variant, namely START-X, which involves using the activation values of $x$ to approximate the saliency of tokens directly. Specifically, for the $i$-th input token $x_i$, we directly compute its saliency value as: $Saliency_X(x_i) = x_i$. With the saliency for each token, we compute the token-level binary mask $\mathcal{M}_S$ as that of START-X and employ Eq. (9) to generate the augmented sequences. In practice, we randomly apply our START method to $50\%$ of the samples in each batch, leaving the remaining samples unperturbed during each training iteration. Our START method is disabled during inference.

In summary, we theoretically investigate the generalization error boundary of Mamba at the token level, highlighting that suppressing domain-related information within input-dependent matrices can effectively reduce the generalization error boundary of the model. Based on the theoretical analysis, we propose the first saliency-driven token-aware transformation for SSMs, designing two different variants for identifying and perturbing the tokens focused on by the input-dependent matrices $B, C, \Delta$. In this way, our method can effectively enhance the reliance of the input-dependent matrices on domain-invariant features and narrow the distance between source and target domains. Notably, our START introduces no additional parameters or inference time, only involving a few matrix operations during training, thus achieving similar linear complexity to VMamba as presented in Appendix A.2.

## 4 Experiments

### 4.1 Experimental Setup

**Datasets.** We perform an extensive evaluation on five DG datasets: **PACS** [24] comprises $9,991$ images of 7 classes from 4 domains: Photo, Art Painting, Cartoon, and Sketch. **OfficeHome** [79] includes $15,588$ images of 65 classes from four diverse domains: Artistic, Clipart, Product, and Real-World, exhibiting a large domain gap. **VLCS** [80] contains $10,729$ images of 5 categories from 4 domains: Pascal, LabelMe, Caltech, and Sun. **TerraIncognita** [81] comprises photographs of wild

Table 2: Performance (%) comparisons with the SOTA DG methods on VLCS and TerraIncognita.

| Method | Params. | Caltech | LabelMe | SUN | PASCAL | Avg. | L100 | L38 | L43 | L46 | Avg. |
|---|---|---|---|---|---|---|---|---|---|---|---|
| | | | | VLCS | | | | TerraIncognita | | | |
| CNN: ResNet-50 | | | | | | | | | | | |
| DeepAll [65] (AAAI'20) | 23M | 97.70 | 64.30 | 73.40 | 74.60 | 77.50 | 49.80 | 42.10 | 56.90 | 35.70 | 46.10 |
| PCL [66] (CVPR'22) | 23M | 99.00 | 63.60 | 73.80 | 75.60 | 78.00 | 58.70 | 46.30 | 60.00 | 43.60 | 52.10 |
| EoA [67] (NeurIPS'22) | 23M | 99.10 | 63.10 | 75.90 | 78.30 | 79.10 | 57.80 | 46.50 | 61.30 | 43.50 | 52.30 |
| EQRM [68] (NeurIPS'22) | 23M | 98.30 | 63.70 | 72.60 | 76.70 | 77.80 | 47.90 | 45.20 | 59.10 | 38.80 | 47.80 |
| SAGM [69] (CVPR'23) | 23M | 99.00 | 65.20 | 75.10 | 80.70 | 80.00 | 54.80 | 41.40 | 57.70 | 41.30 | 48.80 |
| iDAG [70] (ICCV'23) | 23M | 98.10 | 62.70 | 69.90 | 77.10 | 76.90 | 58.70 | 35.10 | 57.50 | 33.00 | 46.10 |
| CCFP [71] (ICCV'23) | 23M | 98.10 | 64.90 | 74.50 | 78.30 | 78.90 | 56.40 | 42.30 | 58.00 | 37.50 | 48.60 |
| PGrad [73] (ICLR'23) | 23M | 98.30 | 64.40 | 74.40 | 79.90 | 79.30 | 51.20 | 43.40 | 60.00 | 41.30 | 49.00 |
| AGFA [74] (ICLR'23) | 23M | **99.00** | 64.50 | 75.40 | 78.90 | 79.50 | 61.00 | 46.20 | 60.30 | 42.30 | 52.40 |
| GMDG [75] (CVPR'24) | 23M | 98.30 | 65.90 | 73.40 | 79.30 | 79.20 | 59.80 | 45.30 | 57.10 | 38.20 | 50.10 |
| ViT-based models | | | | | | | | | | | |
| SDViT [18] (ACCV'22) | 22M | 96.80 | 64.20 | 76.20 | 78.50 | 78.90 | 55.90 | 31.70 | 52.20 | 37.40 | 44.30 |
| GMoE-S [19] (ICLR'23) | 34M | 96.90 | 63.20 | 72.30 | 79.50 | 78.00 | 59.20 | 34.00 | 50.70 | 38.50 | 45.60 |
| SSM-based models | | | | | | | | | | | |
| DGMamba [54] (ACM MM'24) | 22M | 98.90 | 64.30 | **79.20** | 80.80 | 80.80 | 62.70 | 48.30 | 61.10 | 46.40 | 54.60 |
| Strong Baseline [22] | 22M | 97.67 | 64.25 | 75.81 | 79.97 | 79.42±0.25 | 66.39 | 47.27 | 62.42 | 48.56 | 56.16±0.41 |
| START-M (Ours) | 22M | 98.80 | **66.98** | 77.18 | 82.33 | **81.32**±0.33 | 70.13 | **49.98** | 63.02 | **49.49** | 58.16±0.79 |
| START-X (Ours) | 22M | 98.66 | 66.64 | 76.97 | **82.58** | 81.21±0.28 | **70.70** | 49.47 | **63.96** | 48.95 | **58.27**±0.75 |

animals taken by 4 camera-trap domains, with 10 classes and a total of 24, 788 images. **DomainNet** [5] is large-scale with 586, 575 images, having 345 classes from 6 domains, *i.e.*, Clipart, Infograph, Painting, Quickdraw, Real, and Sketch. Results on DomainNet are reported in Appendix A.2.

**Implementation details.** We closely follow the implementation of VMamba [22] and use the VMamba-T, which has similar parameters with ResNet-50 (22M vs. 23M), as the backbone. The backbone is pretrained on the ImageNet [82] for all our experiments. We partition the input image into $4 \times 4$ patches without further flattening the patches into a 1D sequence. The network depth of the VMamba-T backbone is 4 the same as ResNet-50, consisting of 2, 2, 9, and 2 VSS layers, respectively. The embedding dimensions of blocks in the 4 stages are fixed as $[96, 192, 384, 768]$. Following existing DG methods [15, 83], we train the model for 50 epochs using AdamW optimizer and cosine decay schedule, with a batch size of 64, the initial learning rate as $5e - 4$, and the momentum of 0.9. For all experiments, we the ratio $P_{token}$ of augmented tokens to 0.75. We apply the leave-one-domain-out protocol for all benchmarks, where one domain is used for testing, and the remaining domains are employed for training. We select the last-epoch model and report the average accuracy over five runs. All the experiments are run on 4 NVIDIA Teska V100 GPUs.

## 4.2 Main Results

**Evaluation on PACS.** We first compare our method with SOTA CNN-based DG methods on ResNet-50. As shown in Tab. 1, the strong baseline (VMamba) achieves a promising performance, exceeding ResNet-50 by 4.44% (89.94% vs. 85.50%), which indicates its superiority for DG. Moreover, we apply our START to the strong baseline and build advanced models, which can achieve significant improvements without introducing extra parameters. Notably, START-M achieves the SOTA performance, improving baseline by 1.83% (91.77% vs. 89.94%) and yielding the latest CNN-based DG method GMDG [75] by 6.17% (91.77% vs. 85.60%). START-X can also improve the baseline significantly by 1.78% (91.72% vs. 89.94%). Compared with SOTA ViT-based methods, START-M still performs excellent, yielding GMoE-S [19] by 5.07% (91.77% vs. 86.70%) with small network sizes (22M vs. 34M). Finally, our methods beat the recent DGMamba [54], exceeding it by 0.57% (91.77% vs. 91.20%) on average, which proves the effectiveness of our method for DG.

**Evaluation on OfficeHome.** We evaluate the effectiveness of our method on OfficeHome and present the results in Tab. 1. Our methods achieve significant improvements compared with CNN-based methods, *e.g.*, START-M outperforms the SOTA method EoA [67] by 4.59% (77.09% vs. 72.50%) on ResNet-50. Based on the Strong Baseline with high performance, our method can still improve it by 0.66% (77.09% vs. 76.43%). START-M precedes the best MLP-like model ViP-S [78], which learns long-range dependencies along height and weight directions, with a large improvement of 4.71% (77.09% vs. 73.38%). The results justify the superiority of START.

**Evaluation on VLCS.** As presented in Tab. 2, our START achieves the best performance among all competitors, surpassing the top CNN-based method SAGM [69] by 1.32% (81.32% vs. 80.00%).

Additionally, our method significantly improves upon the baseline, outperforming the latest Mamba-based method DGMamba by 0.52% (81.32% vs. 80.80%).

**Evaluation on TerraIncognita.** As shown in Tab. 2, We observe that the VMamba baseline significantly outperforms previous methods, achieving a SOTA performance of 56.16%. Based on the strong baseline, our method can further achieve substantial improvement by 2.11% (58.27% vs. 56.16%), proving that our method effectively suppresses domain-specific features learned by Mamba.

## 4.3 Ablation Study and Analytical Experiments

**Ablation study.** We here validate the effectiveness of each operation in START. Specifically, *w/o Saliency Guided* denotes random selection of tokens for perturbation within input sequences, while *w/o Token Selection* means perturbing the entire input sequences. Tab. 3 presents the results using VMamba backbone on PACS. Both variants show improvements over the baseline, indicating that perturbing style information can mitigate overfitting issues. However, as discussed in Section 3.3,

*w/o Saliency Guided*, which randomly perturbs tokens, fails to provide strong regularization. Besides, the result of *w/o Token Selection* is inferior to our START, suggesting that perturbing background tokens could activate other forms of domain-specific features, potentially hindering model generalization.

Table 3: Ablation study on the PACS dataset.

| Method | Art | Cartoon | Photo | Sketch | Avg. |
|---|---|---|---|---|---|
| Baseline [22] | 91.55 | 85.11 | 99.14 | 83.97 | $89.94_{\pm 0.52}$ |
| w/o. Saliency Guided | 92.11 | 86.23 | 99.10 | 85.82 | $90.81_{\pm 0.24}$ |
| w/o. Token Selection | 92.05 | 86.55 | 98.90 | 86.35 | $90.94_{\pm 0.18}$ |
| START-M (Ours) | **93.29** | **87.56** | 99.14 | 87.07 | $\mathbf{91.77}_{\pm 0.40}$ |
| START-X (Ours) | 92.76 | 87.43 | **99.22** | **87.46** | $91.72_{\pm 0.49}$ |

**Parameter sensitivity.** We explore the sensitivity of our method to the hyper-parameter $P_{token}$, the percentage of perturbed tokens in input sequences. As shown in Fig. 3, START-M consistently performs well across different $P_{token}$ values, demonstrating its effectiveness in perturbing domain-specific information in salient tokens. We notice that START-X, which uses token activation to approximate saliency, performs similarly to START-M when $P_{token}$ is high but is less effective at lower $P_{token}$ values. This indicates differences between attention regions of input-dependent matrices and input sequences. Both methods achieve the highest accuracy at $P_{token} = 0.75$, which is adopted for all experiments.

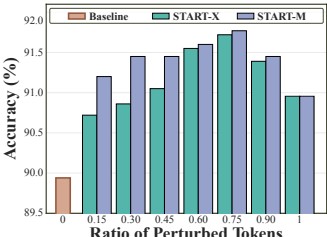

Figure 3: Sensitivity to $P_{token}$.

**Comparisons with other feature identification methods.** We provide comparisons with the "GradCAM" and "Attention Matrix" methods. For the "Grad-CAM" method, we first obtain feature gradients using backpropagation without updating, then compute token saliency and augment salient tokens at each iteration. For the "Attention Matrix" method,

Table 4: Comparison (%) with other salient feature identification methods on PACS with VMamba as the backbone.

| Method | Art | Cartoon | Photo | Sketch | Avg. |
|---|---|---|---|---|---|
| Baseline [22] | 91.55 | 85.11 | 99.14 | 83.97 | $89.94_{\pm 0.52}$ |
| GradCam | 92.56 | 86.99 | 98.98 | 84.92 | $90.86_{\pm 0.24}$ |
| Attention Matrix | 91.75 | 86.68 | 98.88 | 85.76 | $90.77_{\pm 0.30}$ |
| START-M (Ours) | **93.29** | **87.56** | 99.14 | 87.07 | $\mathbf{91.77}_{\pm 0.40}$ |
| START-X (Ours) | 92.76 | 87.43 | **99.22** | **87.46** | $91.72_{\pm 0.49}$ |

since the Mamba architecture lacks explicit attention matrices, we instead use $\alpha$ in Eq. (3) to calculate token saliency. As shown in Tab. 4, on the strong baseline, START still performs much better than these advanced methods, exceeding "GradCAM" by 0.91% (91.77% vs. 90.86%) and "Attention Matrix" by 1.00% (91.77% vs. 90.77%). It is owing to the ability of START to explicitly suppress domain-specific features within input-dependent matrixes.

**Comparisons with other augmentation methods.** We here compare our method with SOTA DG augmentation methods on the VMamba backbone, including MixStyle [13], DSU [14], and ALOFT [15]. As shown in Tab. 5, all the augmentation methods bring performance improvements, indicating that increasing data diversity is beneficial for the generalization ability of Mamba. Notably,

Table 5: Comparisons (%) with SOTA augmentation methods on PACS with VMamba as the backbone.

| Method | Art | Cartoon | Photo | Sketch | Avg. |
|---|---|---|---|---|---|
| Baseline [22] | 91.55 | 85.11 | 99.14 | 83.97 | $89.94_{\pm 0.52}$ |
| MixStyle [13] | 92.05 | 86.55 | 98.90 | 86.35 | $90.94_{\pm 0.18}$ |
| DSU [14] | 92.58 | 85.91 | 98.98 | 85.39 | $90.71_{\pm 0.22}$ |
| ALOFT [15] | 93.07 | 86.04 | 99.16 | 85.31 | $90.89_{\pm 0.24}$ |
| START-M (Ours) | **93.29** | **87.56** | 99.14 | 87.07 | $\mathbf{91.77}_{\pm 0.40}$ |
| START-X (Ours) | 92.76 | 87.43 | **99.22** | **87.46** | $91.72_{\pm 0.49}$ |

START outperforms all the SOTA augmentation methods, *i.e.*, yielding a significant margin of 1.06%

(91.77% vs. 91.72%) from DSU. The results prove the effectiveness of our methods in perturbing domain-specific features within input-dependent matrices.

**Domain gaps in input-dependent matrices.** To verify the effectiveness of our method in reducing domain gaps within input-dependent matrices, we compare domain gaps across different methods using PACS with VMamba. The experiments focus on the last block of VMamba, examining the output feature maps ("Feat.") and the input-dependent

Table 6: Domain gaps within input-dependent matrices.

| Method | $\tilde{\Delta}(\downarrow)$ | $B$ $(\downarrow)$ | $C$ $(\downarrow)$ | Feat. $(\downarrow)$ |
|---|---|---|---|---|
| Baseline [22] | 1.48 | 1.52 | 2.08 | 2.97 |
| MixStyle [13] | 1.73 | 1.36 | 1.90 | 1.91 |
| DSU [14] | 1.38 | 1.28 | 2.18 | 1.59 |
| ALOFT [15] | 1.37 | 1.25 | 2.33 | 1.67 |
| START-M (Ours) | **1.16** | 0.98 | 1.80 | **1.30** |
| START-X (Ours) | 1.23 | **0.91** | **1.52** | 1.37 |

matrices $\tilde{\Delta}$, $B$, and $C$ of the first SS2D. The results in Tab. 6 align well with theoretical analysis in Section 3.2, proving that START effectively reduces domain gaps in the input-dependent matrices.

## 5 Conclusions

In this paper, inspired by the success of Mamba in supervised tasks, we theoretically study the generalizability of Mamba and find that the input-dependent matrices in Mamba could accumulate and amplify domain-specific features during training. To address the issue, we propose a generalized state space model with a saliency-driven token-aware transformation for DG, which can selectively augment domain-specific features within salient tokens focused on by the input-dependent matrices, thus helping the model learn domain-invariant features. Our method outperforms SOTA CNN-based and ViT-based methods by a significant margin with linear complexity and a small-sized network. We hope our work inspires further research in DG and contributes valuable insights to the community.

## 6 Acknowledgment

This work was supported by the National Key R&D Program of China (2023ZD0120700, 2023ZD0120701), NSFC Project (62222604, 62206052), China Postdoctoral Science Foundation (2024M750424), the Fundamental Research Funds for the Central Universities (020214380120), the State Key Laboratory Fund (ZZKT2024A14), the Postdoctoral Fellowship Program of CPSF (GZC20240252), and the Jiangsu Funding Program for Excellent Postdoctoral Talent (2024ZB242).

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

# A Appendix / supplemental material

## A.1 Theoretical Proofs

**Lemma 1 [58].** *Let $\mathcal{F} = \{f \in \mathcal{H}_k : ||f||_{\mathcal{H}_k} \leq 1\}$ denote a function class, where $\mathcal{H}_k$ be a RKHS with its associated kernel $k$. Let $\mathcal{L}_{h,f} : x \to \mathcal{L}[h(x), f(x)]$ be a convex loss-function with a parameter form $|h(x) - f(x)|^q$ for some $q > 0$, and defined $\forall h, f \in \mathcal{F}$, $\mathcal{L}$ obeys the triangle inequality. Let $S$ and $T$ be two samples of size $M$ drawn i.i.d from $D_S$ and $D_T$, respectively. Then, with probability of at least $1 - \delta$ ($\delta \in (0,1)$) for all $h \in \mathcal{F}$, the following holds:*

$$\mathcal{R}_{D_T}[h] \leq \mathcal{R}_{D_S}[h] + d_{\text{MMD}}(D_S, D_T) + \frac{2}{M}(E_{x \sim D_S}[\sqrt{tr(K_{D_S})}] +$$
$$E_{x \sim D_T}[\sqrt{tr(K_{D_T})}]) + 2\sqrt{\frac{\log(\frac{2}{\sigma})}{2M}} + \sigma, \quad (10)$$

*where $d_{\text{MMD}}(D_S, D_T) = \sup_{||f||_{\mathcal{F}_k} \leq 1} \left| \int f d(h(D_S) - h(D_T)) \right|$, $K_{D_S}$ and $K_{D_T}$ are kernel functions computed on samples from $D_S$ and $D_T$, respectively. $\sigma$ is the combined error of the ideal hypothesis $h^*$ on $D_S$ and $D_T$.*

**Theorem 1 (Generalization risk bound).** *With the previous setting and assumptions, let $D_S^i$ and $D_T$ be two sets with $M$ samples independently drawn from $\mathcal{D}_S^n$ and $\mathcal{D}_T$, respectively. For any $\delta \in (0,1)$ with probablity of at least $1 - \delta$, for all $h \in \mathcal{H}$, the following inequality holds:*

$$R_{D_T}(h) \leq \sum_{n=1}^{N} \pi_n R_{D_S}^n(h) + d_{\text{To-MMD}}(D_T, \bar{D}_T) + \sup_{i,j \in [N]} d_{\text{To-MMD}}(D_S^i, D_S^j) + 2\lambda_\pi + \sigma \quad (11)$$

*where $\lambda_\pi = \frac{1}{M}(\sum_{n=1}^{N} \pi_n \mathbb{E}_{x \sim D_S^n}[\sqrt{tr(K_{D_S^n})}] + \mathbb{E}_{x \sim D_T}[\sqrt{tr(K_{D_T})}]) + \sqrt{\frac{log(2/\epsilon)}{2M}}$, and $\sigma$ is the minimum combined error of the ideal hypothesis $h^*$ on both $D_S$ and $D_T$. Let $\gamma_T = d_{\text{To-MMD}}(D_T, \bar{D}_T)$ and $\gamma_S = \sup_{i,j \in [N]} d_{\text{To-MMD}}(D_S^i, D_S^j)$, respectively.*

**Proof.** We initially investigate the relationship between the MMD [58] and To-MMD distances based on **Definition 1** (as presented in Eq. (4)). With the feature extractor $\psi(\cdot)$, we have:

$$d_{\text{MMD}}(D_S, D_T) = \sup_{||f||_{\mathcal{F}_k} \leq 1} \left| \int f d(\psi(D_S) - \psi(D_T)) \right|$$
$$\leq \sup_{||f||_{\mathcal{F}_k} \leq 1} \left| \int f d(\frac{1}{L} \sum_{t=1}^{L} \sup_{\psi_t \in \Psi_t} (\psi_t(D_S) - \psi_t(D_T))) \right| \quad (12)$$
$$= d_{\text{To-MMD}}(D_S, D_T).$$

Then, for a pair of source domain $D_S^n$ and $D_T$, the following inequality holds:

$$d_{\text{To-MMD}}(D_S^n, D_T) \leq d_{\text{To-MMD}}(D_S^n, \bar{D}_T) + d_{\text{To-MMD}}(\bar{D}_T, D_T), \quad (13)$$

with which we can derive the weighted sum of To-MMD between source domains and target domain:

$$\sum_{n=1}^{N} \pi_n d_{\text{To-MMD}}(D_S^n, D_T) \leq \sum_{n=1}^{N} \pi_n d_{\text{To-MMD}}(D_S^n, \bar{D}_T) + d_{\text{To-MMD}}(\bar{D}_T, D_T)$$
$$\leq \sup_{i,j \in [N]} d_{\text{To-MMD}}(D_S^i, D_S^j) + d_{\text{To-MMD}}(\bar{D}_T, D_T). \quad (14)$$

With the above preparations, we now derive the generalization error bound of the Mamba model on the unseen target domain. Recalling that **Lemma 1** indicates the generalization error bound between two different distributions, we generalize it to the scenario of multiple source domains:

$$R_{D_T}(h) \le \sum_{n=1}^{N} \pi_n R_{D_S}^n(h) + \sum_{n=1}^{N} \pi_n d_{\text{To-MMD}}(D_S^n, D_T) +$$

$$2\left(\frac{1}{M}\left(\sum_{n=1}^{N} \pi_n \mathbb{E}_{x \sim D_S^n}\left[\sqrt{tr(K_{D_S^n})}\right] + \mathbb{E}_{x \sim D_T}\left[\sqrt{tr(K_{D_T})}\right]\right) + \sqrt{\frac{\log(2/\epsilon)}{2M}}\right) + \sigma \quad (15)$$

$$\le \sum_{n=1}^{N} \pi_n R_{D_S}^n(h) + d_{\text{To-MMD}}(D_T, \bar{D}_T) + \sup_{i,j \in [N]} d_{\text{To-MMD}}(D_S^i, D_S^j) + 2\lambda_\pi + \sigma,$$

where $\lambda_\pi = \frac{1}{M}\left(\sum_{n=1}^{N} \pi_n \mathbb{E}_{x \sim D_S^n}\left[\sqrt{tr(K_{D_S^n})}\right] + \mathbb{E}_{x \sim D_T}\left[\sqrt{tr(K_{D_T})}\right]\right) + \sqrt{\frac{\log(2/\epsilon)}{2M}}$ and $\sigma$ is the minimum combined error of the ideal hypothesis $h^*$ on both $D_S$ and $D_T$.

**Proposion 1 (Accumulation of Domain Discrepancy).** *Given two distinct domains $D_S$ and $D_T$, the token-level domain distance $d_{\text{To-MMD}}(D_S, D_T)$ depends on $d_{C\tilde{\Delta}Bx}(\bar{x}_i^S, \bar{x}_i^T)$ and $d_{\tilde{\Delta}}(\bar{x}_i^S, \bar{x}_i^T)$ for the i-th token. For the entire recurrent process, domain-specific information encoded in $S_\Delta$, $S_C$, and $S_B$ will accumulate, thereby amplifying domain discrepancy.*

**Proof.** For simplification, we use $\bar{x}^S \in \mathbb{R}^L$ and $\bar{x}^T \in \mathbb{R}^L$ to denote the sample mean embeddings for the $D_S$ and $D_T$, respectively. $L$ represents the token sequence length. To investigate the generalization error boundary of Mamba, we explore a simplified problem in conjunction with a single S6 layer, *i.e.*, $\bar{y} = \alpha \bar{x}$, where $\alpha \in \mathbb{R}^{L \times L}$ is the data-dependent matric. Empirically, based on Eq. (3) and Eq. (4), we estimate the token-level domain gap using Euclidean distance of $\bar{y}^S$ and $\bar{y}^S$,

$$||\bar{y}^S - \bar{y}^T||^2 = \sqrt{\sum_{i=1}^{L}(\bar{y}_i^S - \bar{y}_i^T)^2} = \sqrt{\sum_{i=1}^{L}\left(\sum_{j=1}^{i}(\alpha_{i,j}^s \bar{x}_j^s - \alpha_{i,j}^t \bar{x}_j^t)\right)^2} \quad (16)$$

Combined with Eq. (2) and Eq. (3), we represent $\alpha$ as a direct function of the input $\bar{x}$:

$$\alpha_{i,j} = S_C(\bar{x}_i)\left(\exp\left(\sum_{k=j+1}^{i} \text{softmax}(S_\Delta(\bar{x}_k))A\right)\right)\text{softmax}(S_\Delta(\bar{x}_j))S_B(\bar{x}_j) \quad (17)$$

Since the SSM layer calculates $y_i$ based on the continuous subsequence $[\bar{x}_1, \bar{x}_2, \cdots, \bar{x}_i]$, we here analyze the domain gap of the extracted features at the token level, *i.e.*, $|y_i^S - y_i^T|$. Specifically, assuming that we have calculated $|y_i^S - y_i^T| = \beta$ ($1 \le i < L$), from Eq. (16), we can derive:

$$|y_{i+1}^S - y_{i+1}^T| - |y_i^S - y_i^T| = \left|\sum_{j=1}^{i+1}(\alpha_{i+1,j}^S \bar{x}_j^S - \alpha_{i+1,j}^T \bar{x}_j^T)\right| - \left|\sum_{j=1}^{i}(\alpha_{i,j}^S \bar{x}_j^S - \alpha_{i,j}^T \bar{x}_j^T)\right|$$

$$= \left|\sum_{j=1}^{i}[(\alpha_{i+1,j}^S - \alpha_{i,j}^S)\bar{x}_j^S - (\alpha_{i+1,j}^T - \alpha_{i,j}^T)\bar{x}_j^T] + (\alpha_{i+1,i+1}^S \bar{x}_{i+1}^S - \alpha_{i+1,i+1}^T \bar{x}_{i+1}^T)\right| \quad (18)$$

With the defined $\alpha$ in Eq. (17) and denoting $\text{softmax}(S_\Delta(\cdot))$ as $\tilde{S}_\Delta(\cdot)$ for brevity, we can express:

$$\alpha_{i+1,j} - \alpha_{i,j} = S_C(\bar{x}_{i+1})\left(\exp\left(\sum_{k=j+1}^{i+1} \tilde{S}_\Delta(\bar{x}_{i+1})A\right)\right)\tilde{S}_\Delta(\bar{x}_j)S_B(\bar{x}_j)$$

$$- S_C(\bar{x}_i)\left(\exp\left(\sum_{k=j+1}^{i} \tilde{S}_\Delta(\bar{x}_i)A\right)\right)\tilde{S}_\Delta(\bar{x}_j)S_B(\bar{x}_j) \quad (19)$$

$$= \left[\frac{S_C(\bar{x}_{i+1})}{S_C(\bar{x}_i)}\exp(\tilde{S}_\Delta(\bar{x}_{i+1})A) - 1\right] \cdot \alpha_{i,j}$$

Considering that the differences between adjacent tokens are generally small, $S_C(\bar{x}_{i+1})/S_C(\bar{x}_i)$ could be approximated to 1. When the dimension of $\bar{x}$ is relatively large, then for Eq. (19), we have:

$$\alpha_{i+1,j} - \alpha_{i,j} \approx \tilde{S}_\Delta(\bar{x}_{i+1})A \cdot \alpha_{i,j} \tag{20}$$

Then, we substitute Eq. (20) into Eq. (18) to derive the following formula:

$$
\begin{aligned}
&|y^S_{i+1} - y^T_{i+1}| - |y^S_i - y^T_i| \\
=&|\sum_{j=1}^{i}[\tilde{S}_\Delta(\bar{x}^S_{i+1})A \cdot \alpha_{i,j}\bar{x}^S_j - \tilde{S}_\Delta(\bar{x}^T_{i+1})A \cdot \alpha_{i,j}\bar{x}^T_j] + (\alpha^S_{i+1,i+1}\bar{x}^S_{i+1} - \alpha^T_{i+1,i+1}\bar{x}^T_{i+1})| \\
=&| \left(\tilde{S}_\Delta(\bar{x}^S_{i+1})Ay^S_i - \tilde{S}_\Delta(\bar{x}^T_{i+1})Ay^T_i\right) + \left(\alpha^S_{i+1,i+1}\bar{x}^S_{i+1} - \alpha^T_{i+1,i+1}\bar{x}^T_{i+1}\right)|
\end{aligned} \tag{21}
$$

Finally, recalling that $|y^S_i - y^T_i| = \beta$, we can express $|y^S_{i+1} - y^T_{i+1}|$ as following:

$$
\begin{aligned}
|y^S_{i+1} - y^T_{i+1}| =&| \left(I + \tilde{S}_\Delta(\bar{x}^S_{i+1})A\right)\beta + \underline{\left(\tilde{S}_\Delta(\bar{x}^S_{i+1}) - \tilde{S}_\Delta(\bar{x}^T_{i+1})\right)Ay^T_i} \\
&+ \underline{\left(S_C(\bar{x}^S_{i+1})\tilde{S}_\Delta(\bar{x}^S_{i+1})S_B(\bar{x}^S_{i+1})\bar{x}^S_{i+1} - S_C(\bar{x}^T_{i+1})\tilde{S}_\Delta(\bar{x}^T_{i+1})S_B(\bar{x}^T_{i+1})\bar{x}^T_{i+1}\right)}|
\end{aligned} \tag{22}
$$

Let $d_{C\tilde{\Delta}B\bar{x}}(\bar{x}^S_{i+1}, \bar{x}^T_{i+1}) = S_C(\bar{x}^S_{i+1})\tilde{S}_\Delta(\bar{x}^S_{i+1})S_B(\bar{x}^S_{i+1})\bar{x}^S_{i+1} - S_C(\bar{x}^T_{i+1})\tilde{S}_\Delta(\bar{x}^T_{i+1})S_B(\bar{x}^T_{i+1})\bar{x}^T_{i+1}$, and $d_{\tilde{\Delta}}(\bar{x}^S_{i+1}, \bar{x}^T_{i+1}) = \tilde{S}_\Delta(\bar{x}^S_{i+1}) - \tilde{S}_\Delta(\bar{x}^T_{i+1})$. Then, the above equation reveals that for the input tokens $\bar{x}^S_{i+1}$ and $\bar{x}^T_{i+1}$, *the distance between their extracted features, alongside the gap of historical sequences, primarily depends on* $d_{C\tilde{\Delta}Bx}(\bar{x}^S_{i+1}, \bar{x}^T_{i+1})$ *and* $d_{\tilde{\Delta}}(\bar{x}^S_{i+1}, \bar{x}^T_{i+1})$. Besides, the recurrent process in Eq. (22) could also lead to the accumulation or even enhancement of domain-specific information, *i.e.*, if the model extracts domain-related information from the $i$th token, this part of the information will be retained in the features extracted by the $(i + 1)$-th token. *For the whole recurrent process, domain-related information encoded in $S_\Delta$, $S_C$, and $S_B$ will be accumulated and amplified, which will increase the discrepancy in the features extracted by the model for different domains, thus damaging its generalization ability.* Therefore, to reduce the domain gap, it is imperative to suppress domain-specific information learned by $S_\Delta$, $S_C$, and $S_B$. $\square$

**Proposion 2 (Mitigating Domain Discrepancy Accumulation).** *Perturbing domain-specific features in tokens focused on by $S_\Delta$, $S_C$, and $S_B$ can enhance their learning of domain-invariant features, thus effectively mitigating the accumulation issue in these input-dependent matrices.*

**Proof.** Recalling that in the Mamba mode, $S_\Delta$, $S_C$, and $S_B$ are all linear projection layers, which map the input sequence $\bar{x} \in \mathbb{R}^L$ to the data-dependent matrixes $\Delta$, $C$, and $B$, respectively. Hence, we analyze the influence of tokens in $x$ on these matrixes. Taking the matric $B$ as an example, we explore the simplified problem with $S_B(x) = W_B\bar{x}$, where $W_B \in \mathbb{R}^{L \times N}$ and $N$ denotes the dimension of the hinder state. Inspired by previous works [84, 85], we assume that the input sequence $\bar{x}$ could be decomposed to *domain-specific features* $\bar{x}^I$ and *domain-specific features* $\bar{x}^S$. Then, for the $i$-th token $\bar{x}_i$ in $\bar{x}$, the projection matric $B$ could be denoted as:

$$B_i = S_B(\bar{x}_i) = W_{B_i}\bar{x}_i = [W^I_{B_i}\bar{x}^I_i, W^S_{B_i}\bar{x}^S_i] \tag{23}$$

Previous theoretical works [84, 86] have demonstrated that *when a subset of features is perturbed, its variance would be increased, and the model would be regularized to decrease the weights associated with these features to minimize the prediction loss.* Therefore, by perturbing the domain-specific features $x^S_i$, its corresponding weights $W^S_{B_i}$ would be restricted to 0. Simultaneously, due to minimal changes in the domain-invariant feature $x^I_i$, the learning of its corresponding weights $W^I_{B_i}$ is promoted, thus enhancing $S_B$ learning of domain-invariant features.

Furthermore, given the presence of foreground and background in images [54], different tokens contain varied information. Foreground tokens primarily encode domain-invariant semantic features

Table 7: Performance (%) comparisons with SOTA DG methods on the DomainNet dataset with VMamba as the backbone. The best is **bolded**.

| Method | Params | Clipart | Infograph | Painting | Quickdraw | Real | Sketch | Avg. |
|---|---|---|---|---|---|---|---|---|
| CNN: ResNet-50 | | | | | | | | |
| DeepAll [65] (AAAI'20) | 23M | 63.00 | 21.20 | 50.10 | 13.90 | 63.70 | 52.00 | 44.00 |
| PCL [66] (CVPR'22) | 23M | 67.90 | 24.30 | 55.30 | 15.70 | 66.60 | 56.40 | 47.70 |
| EoA [67] (NeurIPS'22) | 23M | 68.30 | 23.10 | 54.50 | 16.30 | 66.90 | 57.00 | 47.70 |
| EQRM [68] (NeurIPS'22) | 23M | 56.10 | 19.60 | 46.30 | 12.90 | 61.10 | 50.30 | 41.00 |
| SAGM [69] (CVPR'23) | 23M | 64.90 | 21.10 | 51.50 | 14.80 | 64.10 | 53.60 | 45.00 |
| iDAG [70] (ICCV'23) | 23M | 67.90 | 24.20 | 55.00 | 16.40 | 66.10 | 56.90 | 47.70 |
| DomainDrop [60] (ICCV'23) | 23M | 62.40 | 21.00 | 50.50 | 13.80 | 64.60 | 52.40 | 44.10 |
| CCFP [71] (ICCV'23) | 23M | 66.40 | 22.90 | 54.00 | 16.20 | 64.50 | 56.70 | 46.80 |
| PGrad [73] (ICLR'23) | 23M | 57.00 | 18.20 | 48.40 | 13.00 | 60.90 | 48.80 | 41.00 |
| AGFA [74] (ICLR'23) | 23M | 66.70 | 22.90 | 54.00 | 16.70 | 65.90 | 56.30 | 47.10 |
| GMDG [75] (CVPR'24) | 23M | 63.40 | 22.40 | 51.40 | 13.40 | 64.40 | 52.40 | 44.60 |
| ViT-based or MLP-like models | | | | | | | | |
| DoPrompt [17] (arXiv'22) | 86M | 67.70 | 24.60 | 54.90 | 17.50 | 69.60 | 55.20 | 48.30 |
| SDViT [18] (ACCV'22) | 22M | 63.40 | 22.90 | 53.70 | 15.00 | 67.40 | 52.60 | 45.80 |
| SSM-based models | | | | | | | | |
| Strong baseline [22] | 22M | 74.12 | 28.06 | 58.26 | 17.85 | 70.10 | 60.33 | 51.45 |
| START-M (Ours) | 22M | 75.11 | **29.41** | 60.25 | 19.31 | **71.05** | **61.58** | 52.79 |
| START-X (Ours) | 22M | **75.28** | 29.36 | **60.33** | **19.55** | 71.01 | 61.30 | **52.81** |

alongside some domain-specific details. Perturbing the domain-related features in these tokens can effectively enhance the model's learning of domain-invariant features. Conversely, background tokens encompass diverse domain-related spurious features that are challenging to fully extract and perturb. As a result, perturbing a subset of domain-related features in these tokens may inadvertently activate other forms of spurious noise, thereby hindering model generalization. To address this issue, leveraging the hidden attention mechanism of Mamba (as depicted in Eq. (3)), where tokens with high saliency are more likely to belong to the foreground, we propose to perturb domain-specific information solely in the tokens focused on by the input-dependent matrices.

## A.2 Additional Experiments

**Evaluation on DomainNet.** We investigate the effectiveness of our method on the large-scale dataset DomainNet. As shown in Tab. 7, on the challenging benchmark, we find that the strong baseline VMamba can achieve the promising performance of $51.45\%$, proving the superiority of Mamba models on large-scale datasets. Based on the strong baseline, our START can still significantly improve the performance, exceeding the baseline by $1.35\%$ ($52.81\%$ vs $51.45\%$). Besides, compared with CNN-based methods, our method can still achieve significant improvements, *e.g.*, outperforming the latest SOTA method AGFA [74] by $5.71\%$ ($52.81\%$ vs $47.10\%$). Our methods also beat the best ViT-based method DoPrompt [17], yielding it by a large margin of $4.51\%$ ($52.81\%$ vs $48.30\%$). The results prove the effectiveness of our method to help the model learn domain-invariant representation.

**Ablation studies on larger datasets.** We provide the ablation studies on the Officehome and TerraIncognita datasets. As shown in Tab. 8, our methods perform the best among all variants, *e.g.*, on TerraIncognita, our START-X outperforms the variant "*w.o.* Saliency Guided" by $0.97\%$ ($58.27\%$ vs. $57.30\%$) and the variant "*w.o.* Token Selection" by $0.83\%$ ($58.27\%$ vs. $57.44\%$). The results demonstrate the effectiveness of all modules in our START methods.

**Effects on other Mamba-based architectures.** To validate the effectiveness of our START on other Mamba architectures, we conducted experiments on the recent ViM [23], using the ViM-T and ViM-S models with different network sizes. The experiments were performed on the PACS dataset with an initial learning rate of $6.25e-6$ and a weight decay of $1e-8$. As shown in Tab. 9, our method consistently improves performance across the models with different scales, *e.g.*, on ViM-S, START-X outperformed the baseline by $2.03\%$ ($89.41\%$ vs. $87.38\%$), and on ViM-T, START-X exceeded the baseline by $1.66\%$ ($86.74\%$ vs. $85.08\%$). These results demonstrate the generalizability of our method across different Mamba architectures.

**Effects on the ViT architecture.** Recalling that our method is derived from the theoretical analysis that input-dependent matrices in Mamba could accumulate domain-related information during training,

Table 8: Ablation studies on different components of START. The experiments are conducted on large datasets, including OfficeHome and TerraIncognita, with VMamba as the backbone.

| Method | OfficeHome | | | | | TerraIncognita | | | | |
|---|---|---|---|---|---|---|---|---|---|---|
| | Art | Clipart | Product | Real | Avg. | L100 | L38 | L43 | L46 | Avg. |
| Baseline [22] | 75.06 | 60.48 | 84.71 | 85.45 | $76.43_{\pm0.15}$ | 66.39 | 47.27 | 62.42 | 48.56 | $56.16_{\pm0.41}$ |
| w/o. Saliency Guided | 75.12 | 61.06 | 84.91 | 85.42 | $76.63_{\pm0.17}$ | 69.49 | 49.10 | 62.70 | 47.92 | $57.30_{\pm0.07}$ |
| w/o. Token Selection | 75.11 | 61.77 | 84.97 | 85.26 | $76.78_{\pm0.07}$ | 68.97 | 49.19 | 62.87 | 48.74 | $57.44_{\pm0.22}$ |
| START-M (Ours) | 75.15 | 62.04 | **85.31** | **85.84** | **$77.09_{\pm0.16}$** | 70.13 | **49.98** | 63.02 | **49.49** | $58.16_{\pm0.79}$ |
| START-X (Ours) | **75.48** | **62.06** | 85.24 | 85.47 | $77.07_{\pm0.07}$ | **70.70** | 49.47 | **63.96** | 48.95 | **$58.27_{\pm0.75}$** |

Table 9: Effects (%) of our START on the Vim [23] architectures. The experiments are conducted on the PACS dataset with the ViM-T and ViM-S as the backbone, respectively.

| Method | Params. | Art | Cartoon | Photo | Sketch | Avg. |
|---|---|---|---|---|---|---|
| Vim-T [23] | 7M | 88.59 | 80.45 | 98.52 | 72.74 | $85.08_{\pm0.14}$ |
| START-Vim-T-M (Ours) | 7M | 90.01 | 80.90 | 98.25 | 74.22 | $85.85_{\pm0.83}$ |
| START-Vim-T-X (Ours) | 7M | 90.97 | 81.53 | 98.90 | 75.57 | $86.74_{\pm0.67}$ |
| Vim-S [23] | 26M | 90.86 | 80.70 | 99.16 | 78.81 | $87.38_{\pm0.70}$ |
| START-Vim-S-M (Ours) | 26M | 93.77 | 83.79 | **99.46** | 79.45 | $89.12_{\pm0.50}$ |
| START-Vim-S-X (Ours) | 26M | **93.98** | **84.23** | 99.44 | **80.00** | **$89.41_{\pm0.42}$** |

Table 10: Effects (%) of our START on the ViT [87] architecture. The experiments are conducted on the PACS dataset with the DeiT-Small as the backbone.

| Method | Params. | Art | Cartoon | Photo | Sketch | Avg. |
|---|---|---|---|---|---|---|
| DeiT-Small [87] | 22M | 87.55 | 82.16 | 98.45 | 75.24 | $85.85_{\pm0.30}$ |
| START-ViT-M (Ours) | 22M | 88.57 | **83.22** | **98.60** | **77.80** | **$87.05_{\pm0.34}$** |
| START-ViT-X (Ours) | 22M | **88.72** | 83.01 | 98.50 | 76.78 | $86.75_{\pm0.22}$ |

Table 11: Performance (%) of our START under the single-source domain generalization (SDG) setting. The experiments are conducted on the PACS dataset with VMamba as the backbone.

| Method | Art | Cartoon | Photo | Sketch | Avg. |
|---|---|---|---|---|---|
| Strong baseline [22] | 75.10 | 83.29 | 44.92 | 74.76 | $69.52_{\pm0.64}$ |
| START-M (Ours) | 78.08 | 85.44 | **45.02** | **78.03** | **$71.64_{\pm0.23}$** |
| START-X (Ours) | **79.40** | **85.66** | 44.87 | 76.24 | $71.54_{\pm0.15}$ |

our START aims to improve the generalization of the Mamba architecture. Nevertheless, the core concept, adaptively perturbing salient tokens in input-dependent matrices, is also applicable to ViTs. Considering that in ViTs, the attention matrix uses query $Q$ and key $K$, and then multiplied by the original feature $V$ to obtain the final representation. We develop our START-M to START-ViT-M, which calculates token saliency from the input-dependent matrices (*i.e.*, $Q \times K^T$), and START-X to START-ViT-X, which uses the activation value of representation $x$ to approximate saliency. The experiments are conducted on the representative ViT architecture, *i.e.*, DeiT-Small, with the PACS dataset. As shown in Tab.10, 1) on the DeiT-Small baseline, our START re-designed for ViTs still can effectively improve the Baseline by a significant margin, *e.g.*, START-ViT-M outperforms the baseline by 1.20% (87.05% vs. 85.85%). The results prove the effectiveness of our START's variants on ViTs; 2) we notice that the VMamba-T ($22M$ parameters) is a stronger baseline model than the DeiT-Small ($22M$ parameters), exceeding it by a large margin of 4.09% (89.94% vs. 85.85%). The results also reveal the advantage of Mamba architecture to learn domain-invariant token dependencies in compressed state space, and our START can further enhance the generalization ability of Mamba.

**Evaluation on single-source domain generalization tasks.** We here evaluate our method under the single-source-domain generalization setting. As shown in Tab. 11, START-M significantly improves the baseline, outperforming it by 2.12% (71.64% vs. 69.52%). These results prove that our method enhances model generalization by simulating domain shifts through salience-driven token transformation, improving performance in both multi-source and single-source DG tasks.

Table 12: Effectiveness (%) of our START on SOTA augmentation methods. The experiments are conducted on the PACS dataset with VMamba as the backbone.

| Method | Art | Cartoon | Photo | Sketch | Avg. |
|---|---|---|---|---|---|
| Strong baseline [22] | 91.55 | 85.11 | 99.14 | 83.97 | $89.94_{\pm0.52}$ |
| DSU [14] | 92.58 | 85.91 | 98.98 | 85.39 | $90.71_{\pm0.22}$ |
| START-DSU-M (Ours) | 92.38 | 88.18 | 99.22 | **86.26** | $91.51_{\pm0.33}$ |
| START-DSU-X (Ours) | 92.84 | 88.17 | 99.34 | 86.18 | $\mathbf{91.63}_{\pm0.25}$ |
| ALOFT [15] | 93.07 | 86.04 | 99.16 | 85.31 | $90.89_{\pm0.24}$ |
| START-ALOFT-M (Ours) | 93.07 | 88.01 | 99.34 | 85.72 | $91.54_{\pm0.24}$ |
| START-ALOFT-X (Ours) | **93.12** | **88.23** | **99.40** | 85.65 | $\mathbf{91.60}_{\pm0.38}$ |

Table 13: Performance (%) of our START in different layers of the network. The experiments are conducted on the PACS dataset with VMamba as the backbone.

| Method | Art | Cartoon | Photo | Sketch | Avg. |
|---|---|---|---|---|---|
| Baseline [22] | 91.55 | 85.11 | 99.14 | 83.97 | $89.94_{\pm0.52}$ |
| START-M (L1&2) | 92.85 | 87.07 | 98.96 | 85.36 | $91.06_{\pm0.19}$ |
| START-M (L3&4) | 92.77 | 86.50 | 98.66 | 85.85 | $90.95_{\pm0.22}$ |
| START-M (L1&2&3&4) | **93.29** | **87.56** | **99.14** | **87.07** | $\mathbf{91.77}_{\pm0.40}$ |
| START-X (L1&2) | 92.46 | 86.33 | 98.96 | 85.82 | $90.92_{\pm0.20}$ |
| START-X (L3&4) | 92.43 | 85.54 | 99.06 | 86.74 | $90.94_{\pm0.02}$ |
| START-X (L1&2&3&4) | **92.76** | **87.43** | **99.22** | **87.46** | $\mathbf{91.72}_{\pm0.49}$ |
| START-M / X | 92.59 | 87.14 | 98.88 | 85.70 | $91.08_{\pm0.29}$ |

**Effectiveness with other SOTA augmentation methods.** In our method, we utilize a statistics-based style augmentation method to perturb domain-specific information within tokens. We also explore other SOTA DG augmentation methods for comparison, including the DSU [14] that models the distribution of statistics across different samples and resample new statistics from the distribution, and the ALOFT [15] that diversifies the low-frequency spectrum in the frequency domain. These methods primarily perturb style information at the channel level. As shown in Tab. 12, our method significantly improves the performance of the SOTA augmentation methods on the VMamba baseline, *e.g.*, our START-DSU-M achieves a significant improvement over DSU by $0.8\%$ ($91.63\%$ vs. $90.71\%$), exceeding the baseline by $1.69\%$ ($91.63\%$ vs. $89.94\%$). The above results prove that selective perturbation of domain-specific features in salient tokens is crucial for enhancing the generalization capability of Mamba models.

**Effects across different stages.** Our theoretical analysis examined how domain gaps accumulate within each SSM layer. Since one layer's output serves as the next layer's input, domain-specific features from earlier stages increase domain gaps in later stages. To address the issue, we applied START to all layers to reduce domain gaps comprehensively. We also tested START separately in either shallow or deep layers. As shown in Tab. 13, using START in both shallow and deep layers simultaneously performs best, aligning with our theoretical analysis. Applying START-M or START-X randomly across layers also improves performance, though less effectively than using START-M or START-X alone. This may be because START-M and START-X target different domain-related information, leading to incomplete suppression when mixed.

**Computational efficiency.** To evaluate the computational efficiency of our proposed START, we conduct experiments on the PACS dataset and compare our method with existing CNN-based and ViT-based methods. Specifically, we compare the number of parameters, floating point operations per second (FLOPs), the inference times, and the generalization performance of each method. The batch size for evaluating inference time is set to $64$, and the inference time is averaged over $100$ experiments. Since STARR-M and START-X are only activated during training and disabled during inference, they introduce no additional inference time. As shown in Tab. 14, our method has significantly fewer FLOPs than ResNet-50 ($5.68$ vs. $8.26$) while outperforming the DeepAll on ResNet-50 (the baseline that directly trains the model on source domains) by $6.22\%$ ($91.77\%$ vs. $85.50\%$), demonstrating the superiority of our START.

**Visualization explanations.** To provide visual evidence of the effectiveness of our START in suppressing domain-specific features, we use GradCAM [89] to generate attention maps of the last state space layer for both the baseline (pure VMamba) and our START models. As illustrated in Fig. 4,

Table 14: Comparison of the computational efficiency of SOTA DG methods and our START on PACS. The experiments are conducted with $224 \times 224$ image size on one NVIDIA Teska V100 GPU.

| Method | Backbone | Params (M) | GFlops (G) | Time (ms) | Avg. (%) |
|---|---|---|---|---|---|
| DeepAll [65] (AAAI'20) | ResNet-50 | 23 | 8.26 | - | 85.50 |
| iDAG [70] (ICCV'23) | ResNet-50 | 23 | 8.00 | 94 | 88.80 |
| iDAG [70] (ICCV'23) | ResNet-101 | 41 | 15.00 | 495 | 89.20 |
| GMoE-S [19] (ICLR'23) | DeiT-S | 34 | 5.00 | 136 | 88.10 |
| GMoE-B [19] (ICLR'23) | DeiT-B | 133 | 19.00 | 361 | 89.20 |
| ViP [78] (TPAMI'22) | ViP-S | 25 | 13.84 | - | 88.27 |
| GFNet [88] (TPAMI'23) | GFNet-H-Ti | 13 | 4.10 | - | 87.76 |
| DGMamba [54] (ACM MM'24) | VMamba-T | 31 | 5.00 | 233 | 91.20 |
| Strong Baseline [22] | VMamba-T | 22 | 5.68 | 252 | 89.94 |
| START-M (Ours) | VMamba-T | 22 | 5.68 | 252 | 91.77 |
| START-X (Ours) | VMamba-T | 22 | 5.68 | 252 | 91.72 |

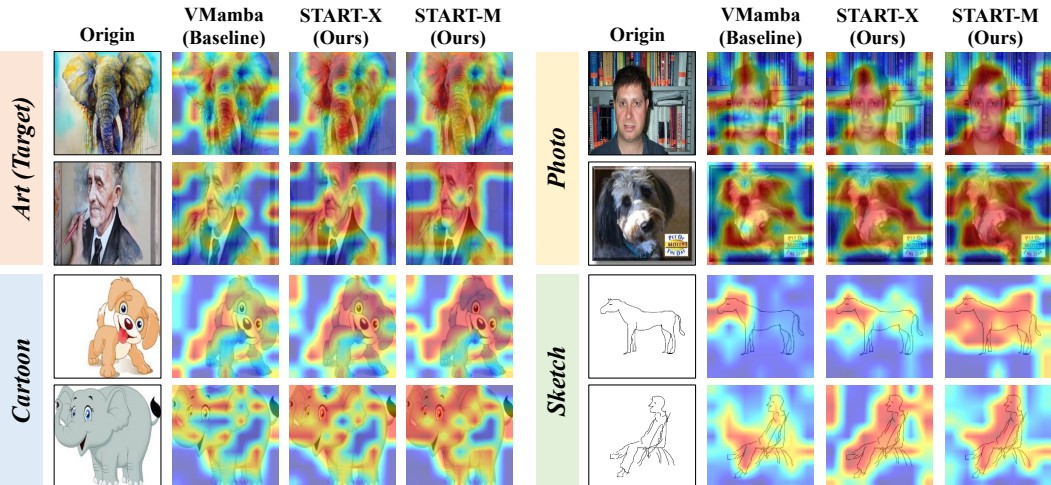

Figure 4: **Visualization results of our START.** The experiments are conducted on the PACS dataset with the "Art" as the target domain. We visualize the attention maps of the last layer in the VMamba backbone. For each sample, the first column is the original image, the second column is the attention map of the baseline (*i.e.*, VMamba), and the third and last columns are the attention maps of our START-X and START-M, respectively. Our methods help the model learn more domain-invariant semantic features, *e.g.*, holistic shape structure, than the pure VMamba baseline.

the VMamba baseline tends to focus on specific local patches that encode domain-specific features, leading to overfitting to source domains. In contrast, our START methods effectively reduce the model's focus on domain-specific features, enabling it to capture generalizable global dependencies of tokens. For instance, in the case of the person image in the Art domain, the baseline focuses on multiple local regions in both the foreground and the background, making the model sensitive to domain shifts and likely to misclassify samples. Conversely, our START models mainly focus on the foreground, specifically the whole face of the person. The results prove the effectiveness of START in learning comprehensive domain-invariant features, making it a promising method for DG tasks.

**Difference from the related work.** In essence, our method significantly differs from DGMamba [54] in their motivations, goals and methods. 1) *Different Motivations*: DGMamba observes that hidden states could amplify domain-related information, and proposes a heuristic method to address the issue. However, it lacks a deep analysis of the phenomenon. Differently, we first theoretically delve into the generalizability of Mamba, revealing how input-dependent matrices contribute to domain gap accumulation. Based on the analysis, we developed START to enhance Mamba's generalization. 2) *Different Goals and Methods*: DGMamba aims to enforce the model to focus on object tokens, perturbing object tokens while replacing context tokens. It ignores that object tokens could be misclassified as context tokens, replacing which would hinder the model from learning semantics. Inversely, START aims to suppress domain-specific information in tokens focused on input-dependent

matrixes, perturbing only styles while keeping contents unchanged. Notably, DGMamba uses the GradCAM [89] for context patch identification, requiring two backpropagations per iteration. Conversely, our START uses input-dependent matrixes to calculate token saliency during forward propagation, needing only one backpropagation and thus reducing training time.

### A.3 Broader Impact

Our work aims to enhance model generalization and computational efficiency, enabling robust performance across diverse domains with varying distributions. By mitigating the overfitting issue on limited source domains and improving performance on unseen target domains, we believe it will have a positive societal impact.

### A.4 Limitations of Our Work

In the paper, we employ style perturbation to perturb domain-specific information within salient tokens of input sequences. However, there are various forms of domain-specific information in the images, which could not be completely suppressed by our method. To address this issue, a potential solution is to design advanced feature disentanglement methods that adaptively distinguish and perturb domain-specific information. Designing class-aware feature augmentation could also alleviate the accumulation of domain-related features. We will explore these solutions in future work.

