# OpenReview forum: "START: A Generalized State Space Model with Saliency-Driven Token-Aware Transformation"
_NeurIPS.cc/2024/Conference — NeurIPS 2024 poster_

### Official Review · Reviewer_4RVV · 2024-07-12

**Soundness:** 4
**Presentation:** 4
**Contribution:** 3
**Rating:** 5
**Confidence:** 4

**Summary:**

This paper considers the domain generalization problem and analyze generalization risk across different domains based on the State Space Model (SSM). At the beginning, the advance of SSM over ViT and CNN is introduced. Specifically, the generalization risk bound is formulated by the token-level maximum mean discrepancy. Then, a detailed theoretical analysis is conducted to indicate the accumulation of domain bias on instance-dependent features across different stages. Then, the saliency is roughly estimated by the variables within SSM and the tokens with high saliency is replaced with instance-independent features to help suppress the domain-related information. Under this merit, there are two variants are designed for token selection (START-X, START-M). Detailed experimental study is also provided.

**Strengths:**

1.	Clear paper writing.
2.	The analysis of SSM on computer vision tasks is important, which provide concrete analysis of domain bias in the network architecture.
3.	Based on the analysis, the salience is an interesting matric in indicating the selection of tokens to mitigate the performance gap across domains.

**Weaknesses:**

1.	I am generally curious about the condition & assumption of this analysis. Specifically, I am not sure how the number of training sources will impact analysis and the performance of derived approach. Intuitively speaking, a model trained on more diverse source domains can perform better on the unseen domains. However, in your Eq. (5), it does not directly indicate that the performances by large N and small N will be different. Necessary explanation should be made.
2.	Selecting and replacing the tokens with high saliency can help mitigate the domain-related information. However, some useful information can also be hurt inevitably. I am wondering how the approaches can be used to deal with this trade-off.
3.	For your experiments, what is the value of N. Have you considered single-source domain generalization in your evaluation, i.e., N=1?
4.	Honestly, the performance gain is not that significant on Office-Home, comparing with PACS. Is there any reason for such trivial gain?
5.	Though theoretical evidence is additionally provided to show the accumulation of domain bias in the instance-dependent features, how your theory can be used to show the relationship between degree of bias by instance-dependent metrics across different stages. To make the approaches derived from your analysis more effective, should a shallower stage or deeper stage be strongly constrained. Further, whether START-M and START-X can be interactively used in different layers/stages?

**Questions:**

Besides the questions I raised in Weakness, I have one more questions for the author about the necessity of this research.

6.	I am generally interested in whether Mamba and its variants are better structures in image processing, comparing with ViT and CNN. If not, what the importance of your paper in analyzing the property of Mamba in domain generalization?

**Limitations:**

No Limitation section is mentioned in the main paper.

I am not sure about the generalization ability of this analysis to different architectures.

---

> ### Author Rebuttal · Authors · 2024-08-06
>
> Thanks for your valuable reviews. Due to the character limit of $6000$ here, the tables are displayed in the ''glocal_tables.pdf''.
>
> ### **Q1: The number of source domains.**
> Thanks for your constructive feedback.
> Following previous works [9,13,66], we consider $N \geq 1$ source domains to theoretically study model generalization. Our analysis and methods apply to varying numbers of source domains $N$.  Indeed, models trained on more diverse source domains (_i.e._, larger $N$) could exhibit better generalization performance. As shown in Eq.($5$) of the paper, $\bar{D}\_T \in \Lambda_S$ denotes the closest domain to target domain $D\_T$ within source domains $D\_S$. Increasing $N$ could bring $\bar{D}\_T$ closer to $D\_T$, reducing the second term in Eq.($5$), $d\_{\\text{To-MMD}}(D\_T, \bar{D}\_T)$. Meanwhile, a larger $N$ helps the model learn domain-invariant features more effectively, reducing the third term, $\sup\_{i, j \in [N]} d_{\\text{To-MMD}}(D_S^i, D_S^j)$. We aim to help the model learn domain-invariant information from a fixed number $N$ of source domains, rather than increasing $N$. Thus, our analysis focuses on the accumulation of feature-level domain gaps during training. In our experiments, we address the DG task with multiple source domains. Following existing DG methods [10,58,63], for a dataset with $K$ domains (e.g., $K=4$ for PACS), we train on $K-1$ domains and use the remaining domain for evaluation. We will include these details in the revised version.
>
> ### **Q2: For useful information in tokens with high saliency.**
> Yes, directly replacing high-saliency tokens could harm valuable information.  To address this, START modifies only the style information of salient tokens to preserve semantics (L$219-221$ of the paper). Besides, we apply START to random $50\\%$ of the samples per iteration, which could prevent excessive noise (L$235-237$ of the paper). These designs generate diverse style perturbations while protecting original semantic features, improving model generalization. We will include this discussion in the revision.
>
>
> ### **Q3: Performance for SDG.**
> In our experiments, we evaluated our method under multiple-source-domains generalization setting, following existing DG methods [10,58,63]. We trained on all but one domain, using the remaining domain as the target. For example, $N=3$ for PACS with $4$ domains and $N = 5$ for DomainNet with $6$ domains. We also evaluated our method under single-source-domain generalization setting. As shown in Tab.$6$ of ''global_tables.pdf'', START-M significantly improves the baseline, outperforming it by $2.12\\%$ ($71.64\\%$ vs. $69.52\\%$). These results prove that our method enhances model generalization by simulating domain shifts through salience-driven token transformation, improving performance in both multi-source and single-source DG tasks.
>
>
> ### **Q4: Discussion about performance on OfficeHome.**
> OfficeHome is more challenging than PACS due to its larger number of categories ($65$ vs. $7$) and samples ($15,588$ vs. $9,991$). Pure VMamba achieves $76.43\\%$ on OfficeHome, surpassing many SOTA DG methods, making further improvements difficult. Nonetheless, our method improves VMamba by $0.66\\%$ ($77.09\\% \pm 0.16\\%$ vs. $76.43\\% \pm 0.15\\%$), averaged over five runs. We tested START on larger datasets (TerraIncognita and DomainNet). As shown in Tab.$1$ of ''global_tables.pdf'', START achieves the best average performance across five datasets, excelling on TerraIncognita and DomainNet. The results confirm our method's ability to stably improve model generalization.
>
> ### **Q5: Effects across different stages.**
> Our theoretical analysis examined how domain gaps accumulate within each SSM layer. Since one layer's output serves as the next layer's input, domain-specific features from earlier stages increases domain gaps in later stages. To address the issue, we applied START to all layers to reduce domain gaps comprehensively. We also tested START in either shallow or deep layers separately. As shown in Tab.$8$  of "global\_tables.pdf", using START in both shallow and deep layers simultaneously performs best, aligning with our theoretical analysis. Applying START-M or START-X randomly across layers also improves performance, though less effectively than using START-M or START-X alone. This may be because START-M and START-X target different domain-related information, leading to incomplete suppression when mixed. We appreciate your feedback and will further investigate START's impact at different stages, exploring adaptive selection strategies for START-M and START-X in future work.
>
>
> ### **Q6: Effects of Mamba and its variants.**
> Yes, we believe Mamba is a strong alternative to ViT and CNN in image processing. Pioneering works have shown Mamba's effectiveness [22,23], and recent studies have confirmed its success across various visual tasks [48,49,50]. Mamba's selective scanning mechanism allows it to capture both global and local token dependencies, potentially combining the advantages of CNNs and ViTs. This inspired our work, which is the first to explore the advantages of Mamba in DG from a theoretical perspective. The experiments prove the superiority of Mamba. We hope our work will encourage further research in the field.
>
> ### **Q7: Limitations of our method.**
> Thanks. The limitations of our method have been discussed in Appendix A.$4$, highlighting challenges in accurately identifying and suppressing domain-specific information. Our analysis of hidden space modeling in the Mamba architecture aims to theoretically assess its generalization capability. Our analysis and method could be transferred to other architectures. As shown in Tab.$5$ of ''global\_table.pdf'', we re-designed our method for ViTs, and preliminary results demonstrate its generalizability. Thanks and we will explore extending our work to other architectures in future research.

---

> > ### Comment · Reviewer_4RVV · 2024-08-14
> > **Response to Author's rebuttal**
> >
> > I would appreciate the authors' effort in this rebuttal and I think my questions have been well-addressed. I am generally positive towards this paper submission.

---

### Official Review · Reviewer_3dvs · 2024-07-15

**Soundness:** 2
**Presentation:** 2
**Contribution:** 2
**Rating:** 5
**Confidence:** 4

**Summary:**

This paper targets adapting Mamba for Domain Generalization. The authors find that find the input-dependent matrices in SSMs could accumulate and amplify domain-specific features, so they hinders model generalization. To address this paper, the authors selectively perturb and suppress domain-specific features in salient tokens within the input-dependent matrices of SSMs. Experiments show that their performance can improve Vmamba on Domain Generalization benchmark.

**Strengths:**

- This paper is early work to adapt Mamba for Domain Generalization.

- The baseline is Vmamba and START can improve VMamba for Domain Generalization benchmark.

**Weaknesses:**

- The only fair baseline is Vmamba (Strong Baseline denoted in the paper). Can other DG methods on CNN or ViT directly transfer to VMamba? If they can, how about their performance? Is your method still the best?

- ViT has attention maps, which are also input-dependent. Can your methods transfer to ViT? Or your method can only be applied to Mamba?

**Questions:**

See weaknesses. Another suggestion is to list model accuracy on ImageNet in Table 1 and Table 2.

**Limitations:**

The authors have discussed limitations in the appendix.

---

> ### Author Rebuttal · Authors · 2024-08-06
>
> ### **Q1: Comparision with CNN-based or ViT-based DG methods on VMamba.**
> Thanks for your feedback.
> Indeed, the previous DG method could be transferred to VMamba. However, these CNN-based or ViT-based DG methods ignore the accumulation of domain gaps in the state space modeling process of Mamba, leading to their insufficient improvement of model generalization.
> To address this issue, we developed a sequence-level saliency-driven token transformation, which aims to suppress domain-specific information within the SSM layers explicitly.
> We compare our method with representative SOTA DG methods on the VMamba backbone, including the CNN-based methods (MixStyle and DSU) and MLP-based method (ALOFT).
> As shown in Tab.$4$ of the ''global\_tables.pdf'', on the strong baseline VMamba, our method achieves the best information compared with previous DG methods, suppressing the CNN-based DSU by $1.06\\%$ ($91.77\\%$ vs. $90.71\\%$) and the MLP-based ALOFT by $0.88\\%$ ($91.77\\%$ vs. $90.89\\%$).
> The results demonstrate that our method effectively mitigates the accumulation of domain differences in the input-dependent matrixes and enhances the model learning of domain-invariant information.
>
>
> | Method | Art | Cartoon | Photo | Sketch | Avg. |
> | :----: | :----: | :----: | :----: | :----: | :----: |
> | Baseline [22] | 91.55 | 85.11 | 99.14 | 83.97 | 89.94 ± 0.52 |
> | MixStyle [13] | 92.05 | 86.55 | 98.90 | 86.35 | 90.94 ± 0.18 |
> | DSU [14] | 92.58 | 85.91 | 98.98 | 85.39 | 90.71 ± 0.22 |
> | ALOFT [15] | 93.07 | 86.04 | 99.16 | 85.31 | 90.89 ± 0.24 |
> | **START-M (Ours)** | **93.29** | **87.56** | 99.14 | 87.07 | **91.77** ± 0.40|
> | **START-X (Ours)** | 92.76 | 87.43 | **99.22** | **87.46** | 91.72 ± 0.49 |
>
>
> ### **Q2: Performance of our START on ViTs.**
> Recalling that our method is derived from the theoretical analysis that input-dependent matrices in Mamba could accumulate domain-related information during training, our START aims to improve the generalization of the Mamba architecture. Nevertheless, the core concept, adaptively perturbing salient tokens in input-dependent matrices, is also applicable to ViTs. Considering that in ViTs, the attention matrix is computed by query $Q$ and key $K$, and then multiplied by the original feature $V$ to obtain the final representation. We develop our START-M to START-ViT-M, which calculates token saliency from the input-dependent matrices (_i.e._, $Q \times K^T$), and START-X to START-ViT-X, which uses the activation value of representation $x$ to approximate saliency. The experiments are conducted on the representative ViT architecture, _i.e._, DeiT-Small, with the PACS dataset. As shown in Tab.$5$ of the ''global\_tables.pdf'', 1) on the DeiT-Small baseline, our START re-designed for ViTs still can effectively improve the Baseline by a significant margin, _e.g._, START-ViT-M outperforms the baseline by $1.20\\%$ ($87.05\\%$ vs. $85.85\\%$). The results prove the effectiveness of our START's variants on ViTs; 2) we notice that the VMamba-T ($22M$ parameters) is a stronger baseline model than the DeiT-Small ($22M$ parameters), exceeding it by a large margin of $4.09\\%$ ($89.94\\%$ vs. $85.85\\%$). The results also reveal the advantage of Mamba architecture to learn domain-invariant token dependencies in compressed state space, and our START can further enhance the generalization ability of Mamba.
>
>
> | Method | Art | Cartoon | Photo | Sketch | Avg. |
> | :----: | :----: | :----: | :----: | :----: | :----: |
> | DeiT-Small | 87.55 | 82.16 | 98.45 | 75.24  | 85.85 ± 0.30   |
> | **START-ViT-M (Ours)** | 88.57 | **83.22** | **98.60** | **77.80** | **87.05** ± 0.34 |
> | **START-ViT-X (Ours)** | **88.72** | 83.01 | 98.50 | 76.78 | 86.75 ± 0.22 |
>
>
> ### **Q3: Model accuracy on ImageNet in Tab.1 and Tab.2.**
> Thanks for your advice. Regarding DG issues, existing works typically use ImageNet pre-trained models and train them on downstream DG standard datasets with large domain gaps (_e.g._, PACS, DomainNet) [1,2,3].
> In Tab.$1$ and $2$ of the paper, we evaluate the model by selecting one domain from the DG dataset as the target domain (_e.g._, Photo) and using the remaining domains (_e.g._, Art, Cartoon, Sketch) as source domains for training. We compute the average model performance across different target domains within the dataset.
>
> Since the category space changes after training, models trained on DG datasets cannot be directly evaluated on ImageNet. Therefore, for the trained models on DG datasets, we select similar categories from the ImageNet1K Validation Set for evaluation. Specifically, for models trained on PACS, we selected $6$ similar categories from the ImageNet1K Validation Set, excluding "giraffe," which has no corresponding category in ImageNet1K. The experiments were conducted on models trained on PACS with "Art" as the target domain.
>
> As shown in the table below, the trained VMamba model achieves an improvement over the pre-trained version, indicating that training on DG datasets with distribution shifts helps the model learn semantic information. Furthermore, both our START-M and START-X methods further improve the accuracy of VMamba by $4.00\\%$ ($90.33\\%$ vs. $86.33\\%$) and  $3.34\\%$ ($89.67\\%$ vs. $86.33\\%$), respectively. These results demonstrate the effectiveness of our methods in reducing the accumulation of domain-specific information and promoting the learning of domain-invariant semantic information.
> We appreciate your suggestion and will incorporate this discussion into the revised paper for improved clarity.
>
> | Method | Pretrained VMamba | Trained VMamba | START-M | START-X |
> | :----: | :----: | :----: | :----: | :----: |
> | ACC. | 73.33 | 86.33 | **90.33** | 89.67 |
>
> [1] Mixstyle neural networks for domain generalization and adaptation. IJCV 2024
>
> [2] Rethinking multi-domain generalization with a general learning objective. CVPR 2024
>
> [3] Dgmamba: Domain generalization via generalized state space model. ACM MM 2024

---

> > ### Comment · Reviewer_3dvs · 2024-08-12
> >
> > Thanks for the rebuttal. After reading the rebuttal, I decided to improve my rating to 5.

---

### Official Review · Reviewer_qvXD · 2024-07-16

**Soundness:** 3
**Presentation:** 3
**Contribution:** 3
**Rating:** 6
**Confidence:** 3

**Summary:**

Advancements in state space models (SSMs), particularly a model called Mamba, have demonstrated efficient performance in supervised learning, offering linear complexity in training and rapid computation during inference, similar to RNNs. This paper explores the potential of the Mamba model for DG and identifies a challenge: input-dependent matrices in SSMs can exacerbate domain-specific features, thus impairing generalization.

To overcome this limitation, the authors introduce a novel SSM-based architecture featuring a saliency-based token-aware transformation, termed START. This architecture effectively reduces domain discrepancies by selectively altering and suppressing domain-specific features within the salient tokens of input-dependent matrices. START achieves state-of-the-art performance in DG, presenting a viable and competitive alternative to both CNNs and ViTs.

**Strengths:**

1. This passage discusses a theoretical analysis of the Mamba model's performance in domain generalization. The analysis reveals that the Mamba model's input-dependent matrices have a tendency to accumulate domain-specific features during their recurrent processing. This accumulation could potentially limit the model’s ability to generalize effectively across different, unseen domains. Essentially, the matrices that should help the model adapt to new data instead make it more sensitive to the specific characteristics of the data it was trained on, thereby impeding its general applicability.

2. This statement outlines the proposal of a new state space model (SSM)-based architecture, which incorporates a saliency-driven, token-aware transformation aimed at addressing the issues previously identified with the Mamba model in domain generalization tasks. This novel architecture is designed to serve as an effective alternative to both convolutional neural networks (CNNs) and vision transformers (ViTs). It boasts an ability to generalize excellently across diverse domains while maintaining the efficient linear complexity characteristic of state space models, making it a promising option for handling domain shifts in a computationally efficient manner.

**Weaknesses:**

It would be better to report computation costs like inference time.

**Questions:**

Please refer to the weakness.

**Limitations:**

This paper discusses the limitations.

---

> ### Author Rebuttal · Authors · 2024-08-06
>
> ### **Q1: Computational costs including inference time.**
> Thanks for your valuable advice. We have provided a comparison of inference times. The batch size for evaluating inference time is set to $64$, and the inference time is averaged over $100$ experiments. Since STARR-M and START-X are only activated during training and disabled during inference, they introduce no additional inference time. As shown in Tab.$7$ of the ''global\_tables.pdf'' and the table below, our methods have computational costs comparable to existing SOTA methods on CNN or ViT while achieving the highest generalization performance, demonstrating the effectiveness of our methods. Thanks and we will add the results and discussions into the revised version of the paper.
>
> | Method | Backbone | Params (M) | GFlops (G) | Time (ms) | Avg. (%) |
> | :----: | :----: | :----: | :----: | :----: | :----: |
> | DeepAll [64] | ResNet-$50 | 23 |8.26 |- |85.50|
> | iDAG [69]  | ResNet-50 |23 |8.00 |94 |88.80|
> | iDAG [69]  | ResNet-101 |41 |15.00  |495 |89.20|
> | GMoE-S [19]  | DeiT-S |34 |5.00 |136 |88.10|
> | GMoE-B [19]  | DeiT-B |133 |19.00 |361 |89.20|
> | ViP [77] | ViP-S |25 |13.84 |- |88.27|
> | GFNet [87] | GFNet-H-Ti |13 |4.10 |- |87.76|
> | DGMamba [78]  | VMamba-T |31 |5.00 |233 |91.20|
> | Strong Baseline [22] | VMamba-T |22 |5.68 |252 |89.94|
> | **START-M (Ours)** | VMamba-T |22 |5.68 |252 |91.77|
> | **START-X (Ours)** | VMamba-T |22 |5.68 |252 |91.72|

---

### Official Review · Reviewer_AMLN · 2024-07-30

**Soundness:** 3
**Presentation:** 3
**Contribution:** 3
**Rating:** 6
**Confidence:** 4

**Summary:**

This paper studies the role of MAMBA architectures on Domain Generalization benchmarks and adapts the architecture to achieve robust generalization.
The motivation to use MAMBA-style architectures is their linear complexity. The authors theoretically analyze conventional MAMBA for DG and make an important finding that internal hidden states in MAMBA capture domain-specific characteristics, thus hurting their generalization capabilities.
To address this, they propose two techniques grounded in theory. Both of them involve suppressing the domain-specific information from salient tokens. These tokens are obtained by looking at the "attention matrix" of the last layer and determining the top K tokens which have high attention values. To "debias" these, the first method mixes these tokens (derived through a mask) with style computed from another image. This way, while the content is preserved, the style is augmented, thus making the network learn domain-invariant features.
In the second method, they calculate the salient tokens by directly relying on the activation values and employ the same procedure as above to produce a new style image.
They have conducted experimental analysis on standard DG benchmarks and demonstrate empirical gains compared to the baselines.

**Strengths:**

This paper comprehensively studies the generalization capabilities of MAMBA (state-space) models with respect to their ability to handle distribution shifts. The theoretical analysis is somewhat straightforward, relying on the MMD measure, but it is an important contribution to the field. The experiments follow standard protocols. The proposed methods are mathematically grounded and produce empirical gains compared to baselines.

**Weaknesses:**

The main weakness is a rather weak motivation for using MAMBA for DG. It almost feels like a low-hanging fruit contribution without proper motivation for the same.
There is another very similar work on using MAMBA for DG called DGMAMBA that was recently published. I commend the authors for including that paper in their experimental comparison. But upon reading that paper, there are very similar ideas being proposed, especially the observation that hidden states of MAMBA carry domain-specific information.
In the author response phase, I would like the authors to clearly differentiate how the proposed approach is different from DG MAMBA. Furthermore, DGMAMBA and the proposed approach produce very similar numbers on at least two datasets.
Second, does the salient token perturbation of both M and X happen from epoch 1? Or is it after a few epochs of training? Since the method relies on ImageNet pre-trained models, maybe from epoch 1, the salient tokens can be meaningfully predicted. But what if the models are not pre-trained? How does the method change?
Small improvement suggestion: It would be nice to remove the overloading of $h$ - in section 3.1 it represents hidden states and in section 3.2 it denotes hypothesis.

**Questions:**

please see weaknesses

**Limitations:**

The authors have sufficiently addressed the limitations.

---

> ### Author Rebuttal · Authors · 2024-08-06
>
> ### **Q1: Motivation of our method.**
> Thanks for your feedback. Some pioneering work have demonstrated the effectivenes of Mamba architecture on various supervised visual tasks. However, few works have studied the generalization ability of Mamba under distribution shift, especially the problem that how to construct a effective Mamba-based model for DG. It motivated us to investigate Mamba's generalization from a theoretical perspective, in which we reveal the issue that the input-dependent matrixes inevitably accumulate domain-specific information and hinder model generalization during training. Based on the theoretical analysis, we propose a saliency-driven token transformation for Mamba, with which we built a generalizable model as a competitive alternative to CNNs and ViTs for DG.
>
>
> ### **Q2: Difference from DGMamba.**
> In essence, our approach significantly differs from DGMamba in motivations, goals and methods.
>
> 1) **Different Motivations:** DGMamba was motivated by the observation that hidden states could amplify domain-related information, and proposes a heuristic method to address the issue. However, it lacks a deep analysis of the phenomenon and fails to reveal its relationship with the model's generalization ability. In contrast, our method is the first to theoretically study the accumulation of domain gaps, revealing that input-dependent matrices could learn and accumulate domain-specific features. Based on the theoretical analysis, we develop START to enhance Mamba's generalization. As shown in Tab.$5$ of the paper and the table below, we also quantitatively validate the effectiveness of our method in reducing domain gaps.
>
> | Method | Domain gap of $\tilde{\Delta} (\downarrow)$ | Domain gap of B $(\downarrow)$ | Domain gap of C $(\downarrow)$ | Domain gap of Feat. $(\downarrow)$|
> | :----: | :----: | :----: | :----: | :----: |
> | Baseline | 1.48 | 1.52 | 2.08 | 2.97 |
> | MixStyle  | 1.73 | 1.36 | 1.90 | 1.91 |
> | DSU | 1.38 | 1.28 | 2.18 | 1.59 |
> | ALOFT | 1.37 | 1.25 | 2.33 | 1.67 |
> | START-M (Ours) | **1.16** | 0.98 | 1.80 | **1.30** |
> | START-X (Ours) | 1.23 | **0.91** | **1.52** | 1.37 |
>
> 2) **Different Goals and Methods:** _DGMamba aims to enforce the model to focus on object tokens_, which is achieved by perturbing the object tokens while shuffling and replacing the context tokens. However, this strategy ignores that object tokens could be misclassified as context tokens, which is likely to damage key semantic information when some object tokens are replaced. Besides, the HSS module of DGMamba aims to mask and suppress features that are lowly activated in hidden states, ignoring that the remaining hidden states could also contain domain-related information. Differently, _our method aims to suppress domain-specific information in tokens focused on by input-dependent matrixes_, which perturbs style information while keeping semantic content unchanged. In this way, our START could effectively mitigate the domain gap accumulation issue in Mamba, thus sufficiently improving generalizability. Notably, DGMamba uses GradCAM for context patch identification, requiring two backpropagations per iteration. Conversely, our START uses input-dependent matrixes to calculate token saliency during forward propagation, needing only one backpropagation and thus reducing training time.
>
> We appreciate your feedback and will include this discussion in the revised version.
>
> ### **Q3: Experimental comparison with DGMamba.**
> As shown in Tab.$1$ of the ''global\_tables.pdf'' and the table below, we compared our method to DGMamba across five DG datasets. Since the code of DGMamba has not been open-source, we report the results from the DGMamba paper.
> The results indicate that our method can stably outperform DGMamba on all five datasets, achieving an average performance improvement of $1.59\\%$ ($72.23\\%$ vs. $70.64\\%$), thereby proving its effectiveness.
> Specifically, on the large-scale datasets TerraIncognita and DomainNet, our method shows substantial gains over DGMamba, _e.g._, START-M outperforms DGMamba by $3.56\\%$ ($58.16\\%$ vs. $54.60\\%$) on TerraIncognita and $3.19\\%$ ($52.79\\%$ vs. $49.60\\%$) on DomainNet.
> On OfficeHome, our results are similar to DGMamba, likely because VMamba is a strong baseline that surpasses existing SOTA DG methods, and the performance on OfficeHome is nearly saturated.
> More advanced methods should be studied to improve model performance on OfficeHome further, and combining our START with DGMamba could be a promising direction.
> We will add this discussion in the revised version.
>
> | Method | VLCS | PACS | OfficeHome | TerraInc | DomainNet | Avg. |
> | :----: | :----: | :----: | :----: | :----: | :----: | :----: |
> | DGMamba | 80.80 | 91.20 | 77.00 | 54.60 | 49.60 | 70.64 |
> | **START-M (Ours)** | **81.32** |  **91.77** |  **77.09** | 58.16 | 52.79 |  **72.23** |
> | **START-X (Ours)** | 81.21 | 91.72 | 77.07 |  **58.27** |  **52.81** | 72.22 |
>
>
>
> ### **Q4: When do START-M and START-X happen?**
> Yes, both START-M and START-X methods are used from the first epoch. In DG setting, existing methods are trained on DG datasets using ImageNet pre-trained models [1,2,3,4].
> If the model is not pre-trained, we recommend first training the model with a warm-up period for a few epochs, and then applying our START to mitigate overfitting to the source domain.
> Thanks for your question. We will explore this direction in future work.
>
> [1] Mixstyle neural networks for domain generalization and adaptation. IJCV 2024
>
> [2] Rethinking multi-domain generalization with a general learning objective. CVPR 2024
>
> [3] Dgmamba: Domain generalization via generalized state space model. ACM MM 2024
>
> [4] MADG: Margin-based Adversarial Learning for Domain Generalization. NeurIPS 2023.
>
> ### **Q5: Symbol improvements.**
> Thanks for your suggestion, we will remove the overloading of $h$ in the revised version and recheck all symbols to improve the readability of the paper.

---

> > ### Comment · Reviewer_AMLN · 2024-08-10
> >
> > Thank you for your insightful discussion. While I am still not convinced by the motivation, I see a strong value in the theoretical analysis and impressive empirical performance. Owing to these, I am increasing my score.
> >
> > I thank the authors for especially running experiments in short notice. I saw the rebuttal pdf and they have provided a lot more additional analysis which needs to be acknowledged.

---

### Official Review · Reviewer_EPCk · 2024-07-30

**Soundness:** 3
**Presentation:** 2
**Contribution:** 2
**Rating:** 5
**Confidence:** 3

**Summary:**

This paper aims to enhance the generalization ability of state space models (SSMs), i.e. Mamba. This paper first provides a theoretical investigation on the generalization ability of the Mamba model under domain shifts and finds that input-dependent matrices within SSMs could accumulate and amplify domain-specific features. Then this paper proposes saliency-based token-aware transformation (START) to selectively augment the domain-related features in salient tokens within the input-dependent matrices of SSMs. The authors also conduct extensive experiments to demonstrate better performance.

**Strengths:**

- This paper examines the generalization ability of SSMs (Mamba), a topic that few studies have explored. This aspect is particularly important if Vision Mamba is to be widely adopted.
- The generalization problem of Mamba and the subsequent solution, i.e. START, is motivated and supported by theoretical analysis.
- START utilizes the input-dependent matrices to identify salient tokens that are related to domains, which is novel. Then, START uses mix-style to augment domain-related features to force the model to learn domain-invariance features, which is tailored to enhance SSMs' generalization ability and has proven effective compared to the strong baseline in the ablation study.

**Weaknesses:**

- Regarding the writing, I would suggest revising Proposions 1 and 2 to be more concise. Rather than using words like "it's essential for $S_\Delta, S_C, S_B$ to xxx" or "$S_\Delta, S_C, S_B$ plays a play crucial role in this context", maybe elaborate exactly how $S_\Delta, S_C, S_B$  affects the generalization ability of SSMs or how it relates to Domain Discrepancy Accumulation in the Proposions, so readers don't have to go to Appendix for details.
- The main theoretical results are a bit confusing and isolated. I don't see how Theorem 1 is related to Equation 6. Specifically, how $\Vert \bar y^S - \bar y^T \Vert^2$ relates to RHS of Equation 5. Maybe a new generalization bound tailored specifically to SSMs can be derived?
- Performance gain over previous DGMamba is marginal on three out of the four datasets. For instance, START only surpasses DGMamba by 0.09\%, 0.5\%, and 0.52\% on PACS, OfficeHome, and VLCS while the standard deviation is even larger than the gain (e.g. 0.4 on PACS and 0.16 on OfficeHome), making these improvements unconvincing.
- In essence, START identifies a salient mask, which is later used to augment the feature. However, START only compares with random and full masking. Some more advanced salient feature identification methods or Interpretability methods, e.g. gradcam or salient mask w/ attention matrix, should be compared to further demonstrate the effectiveness of START.
- Some typos. For example, line 219, xaug should be $x_{aug}$

Despite lacking some comparison, I believe this paper contains novelty and is technically sound. I will raise the point if the above questions are addressed.

**Questions:**

- Can you discuss the DGMamba in related work, given this is the only work that previously explores the DG setting of Vision Mamba and demonstrates strong performance?
- I also suggest using larger datasets to run the ablation study, for instance, Office-Home, TerraIncognita or DomainNet. PACS is a relatively small dataset on which most methods achieve high enough results. Thus, improvements on PACS are generally marginal and not as convincing as the other datasets.
- START builds upon the idea that Input-Dependent Matrices focus on domain-related features. However, it's inevitable that it also focuses on some domain-invariance features. Does this pose a negative effect on the model? Or using, for example, class-aware feature augmentation be better?

**Limitations:**

The authors have addressed the limitations and, if applicable, the potential negative societal impact of their work.

---

> ### Author Rebuttal · Authors · 2024-08-06
>
> Thanks for your valuable reviews. Due to the character limit of $6000$ here, the tables are displayed in ''glocal_tables.pdf''.
>
> ### **Q1: Revisions of Proposition 1 and 2.**
> Thanks. We have modified Propositions 1 and 2 to be more clear and detailed:
>
> **Proposion 1 (Accumulation of Domain Discrepancy).** _Given two domains $D_S$ and $D_T$, token-level domain distance $d\_{\\text{To-MMD}}(D\_S, D\_T)$ depends on $d\_{C \\tilde{\\Delta} B x}(\\bar{x}\_{i}\^S, \\bar{x}\_{i}\^T)$ and $d\_{\\tilde{\\Delta}}(\\bar{x}\_{i}\^S, \\bar{x}\_{i}\^T)$ for the $i$-th token. For the entire recurrent process, domain-specific information encoded in $S\_\\Delta$, $S\_C$, and $S\_B$ will accumulate, thereby amplifying domain discrepancy._
>
> **Proposition 2 (Mitigating Domain Discrepancy Accumulation).** _Perturbing domain-specific features in tokens focused on by $S\_\\Delta$, $S\_C$, and $S\_B$ can enhance their learning of domain-invariant features, thus effectively mitigating the accumulation issue in these input-dependent matrices._
>
>
> ### **Q2: About theoretical results.**
> As shown in Eq.($5$) of Theorem $1$, generalization error bound of Mamba depends on $\\kappa_T = d\_{\\text{To-MMD}}(D\_T, \\bar{D}\_T)$ and $\\kappa_S = \\sup\_{i, j \in [N]} d\_{\\text{To-MMD}}(D\_S\^i, D\_S\^j)$.
> The smaller the two terms, the lower the upper bound of generalization error.
> Following [25,57,60], Gaussian kernel could estimate $\kappa_T$ and $\kappa_S$. Given $\bar{y}^S$ and $\bar{y}^T$ as averaged feature maps from $D_S$ and $D_T$, domain distance is formulated as $k(\bar{y}^S, \bar{y}^T) = \exp(-||\bar{y}^S - \bar{y}^T||^2 / \gamma)$, where $\gamma$ is the kernel parameter.
> We analyze the effects of input-dependent matrices on $||\bar{y}^S - \bar{y}^T||^2$, which is applicable to both $\kappa_T$ and $\kappa_S$.
> As proven in Propositions 1 and 2, mitigating the accumulation of domain-specific information in input-dependent matrices reduces domain gaps, thus lowering generalization error bound.
>
> ### **Q3: Compared with DGMamba on five DG datasets.**
> We report performances of DGMamba from the paper [78] as its codes have not been open-sourced. As shown in Tab.$1$ of ''global\_tables.pdf'', START consistently outperforms DGMamba across all datasets, exceeding it by $1.59\\%$ ($72.23\\%$ vs. $70.64\\%$) on average. On relatively small datasets (PACS, VLCS, and OfficeHome), START achieves modest but stable improvements from DGMamba. The reason might be that performance on these tasks tends to be saturated, while our method still enhances generalization. On larger and more challenging datasets (TerraIncognita and DomainNet), our method outperforms DGMamba largely, _e.g._, surpassing it by $3.19\\%$ ($52.79\\%$ vs. $49.60\\%$) on DomainNet. The improvements stem from START's focus on mitigating domain gap accumulation in input-dependent matrixes of all layers, effectively enforcing the model to capture semantics and sufficiently improving its ability to handle large datasets.
>
> ### **Q4: Compared with other feature identification methods.**
> We compare the "GradCAM" and "Attention Matrix" methods. For "GradCAM," we first obtain feature gradients using backpropagation without updating, then compute token saliency and augment salient tokens at each iteration. For "Attention Matrix," since Mamba lacks explicit attention matrices, we use $\alpha$ in Eq.($3$) of the paper to calculate token saliency. As shown in Tab.$3$ of ''global\_tables.pdf'', on the strong baseline, START still performs much better than these advanced methods, exceeding "GradCAM" by $0.91\\%$ ($91.77\\%$ vs. $90.86\\%$) and "Attention Matrix" by $1.00\\%$ ($91.77\\%$ vs. $90.77\\%$), owing to the ability of START to explicitly suppress domain-specific features within input-dependent matrixes.
>
>
> ### **Q5: Some Typos.**
> Thanks. We will re-check the paper and fix all typos in the revision.
>
> ### **Q6: Differences from DGMamba.**
> Our START differs greatly from DGMamba. 1) **Motivations.** DGMamba observes that hidden states could amplify domain-related information, and proposes a heuristic method to address the issue. However, it lacks a deep analysis of the phenomenon. Differently, we first theoretically delve into the generalizability of Mamba, revealing how input-dependent matrices contribute to domain gap accumulation. Based on the analysis, we developed START to enhance Mamba's generalization. 2) **Goals and Methods.** DGMamba aims to enforce model to focus on object tokens, perturbing object tokens while replacing context tokens. It ignores that object tokens could be misclassified as context tokens, replacing which would hinder the model from learning semantics. Inversely, START aims to suppress domain-specific information in tokens focused on by input-dependent matrixes, perturbing only styles while keeping contents unchanged. We will add this discussion to Related Work.
>
> ### **Q7: Ablation studies on larger datasets.**
> We provide ablation studies on Officehome and TerraIncognita. As shown in Tab.$2$ of the ''global\_tables.pdf'', our methods perform the best among all variants, _e.g._, on TerraIncognita, outperforming the variant _w.o._ Saliency Guided by $0.97\\%$, proving the effectiveness of all modules.
>
> ### **Q8: For domain-invariant features in input-dependent matrices.**
> Thanks. START is based on the idea that Input-Dependent Matrices could learn and accumulate domain-related features, which hinders them from learning domain-invariant features. To resist this challenge, thanks to our START, the domain-invariant features are preserved and enhanced by only perturbing style information while keeping semantics unchanged (as in L$219$-L$228$ of the paper). In this way, the negative effect of perturbation on the model could be effectively mitigated. We also agree that "class-aware feature augmentation" could alleviate the issue, which would be one of the possible directions for our method in future work. The discussion will be added to the revision.

---

> ### Comment · Reviewer_EPCk · 2024-08-09
> **Discussion regarding Q8**
>
> Dear authors,
>
> Thanks for your rebuttal.
>
> Is there a chance that Input-Dependent Matrices could also learn and accumulate domain-invariant features? Despite being harder, this is totally possible because there is no explicit regularization that forces it to learn only domain-related features. In that case, augmenting the invariant features would somewhat lead to degenerated performance.

---

> > ### Author Response · Authors · 2024-08-09
> > **Discussion about Q8: Domain-invariant features in input-dependent matrices.**
> >
> > Dear Reviewer:
> >
> > Thanks for your constructive comment.
> > Yes. As presented in L$582-586$ of Appendix and Eq.(22), _the domain distance between the tokens $\\bar{x}\_{i+1}^S$ and $\\bar{x}\_{i+1}^T$ depends on both the features extracted from these tokens and the domain distances accumulated in the historical sequence._
> > Therefore, the Input-Dependent matrix could also learn and accumulate domain-invariant features, contributing to Mamba’s strong generalization ability, as proved in our experiments (Tab.1 and Tab.2 of the paper).
> >
> > However, a major challenge in DG is the overfitting issue to the limited source domains [1,2,3,4], which makes the model easily mistake domain-specific features for domain-invariant features.
> > Despite Mamba's strong learning capacity, it still suffers from the overfitting issue on DG tasks, inevitably accumulating domain-specific information during training.
> > As presented in Tab.5 of the paper or the table below, the strong baseline, _i.e._, VMamba directly trained on source domains, still exhibits notable domain gaps.
> > Thus, it is crucial to enforce the model to learn domain-invariant features while suppressing domain-specific ones.
> >
> > | Method | Baseline | MixStyle | DSU | ALOFT | START-M (Ours) | START-X (Ours) |
> > | :----: | :----: | :----: | :----: | :----: | :----: | :----: |
> > Domain Gap $\downarrow$ | 2.97 | 1.91 | 1.59 | 1.67 | **1.30** | 1.37 |
> >
> > Nevertheless, It is difficult to directly distinguish and extract domain-invariant features, while mistakenly augmenting these features would somewhat degrade performance.
> > To address the challenge, rather than roughly augmenting all features in salient tokens, our method perturbs only a part of the features, _i.e._, the style statistics that are more likely to contain domain-specific information [5,6,7,8], to reduce the risk of augmenting domain-invariant features.
> > In this way, the model is compelled to mine category-related semantics from the unchanged features, leading to more accurate learning and accumulation of domain-invariant information.
> >
> > Thanks for your insightful comment. We look forward to further discussion if you have any additional questions.
> >
> >
> > [1] Kaiyang Zhou, Ziwei Liu, Yu Qiao, Tao Xiang, and Chen Change Loy. Domain generalization: A survey. TPAMI, 2022.
> >
> > [2] Jindong Wang, Cuiling Lan, Chang Liu, Yidong Ouyang, Tao Qin, Wang Lu, Yiqiang Chen, Wenjun Zeng, and S Yu Philip. Generalizing to unseen domains: A survey on domain generalization. TKDE, 2022.
> >
> > [3] Jintao Guo, Lei Qi, and Yinghuan Shi. Domaindrop: Suppressing domain-sensitive channels for domain generalization. In ICCV, 2023.
> >
> > [4] Yu Ding, Lei Wang, Bin Liang, Shuming Liang, Yang Wang, and Fang Chen. Domain generalization by learning and removing domain-specific features. In NeurIPS, 2022.
> >
> > [5] Kaiyang Zhou, Yongxin Yang, Yu Qiao, and Tao Xiang. Mixstyle neural networks for domain generalization and adaptation. IJCV, 2024.
> >
> > [6] Chenming Li, Daoan Zhang, Wenjian Huang, and Jianguo Zhang. Cross contrasting feature perturbation for domain generalization. In ICCV, 2023.
> >
> > [7] Sangrok Lee, Jongseong Bae, Ha Young Kim. Decompose, Adjust, Compose: Effective Normalization by Playing with Frequency for Domain Generalization. CVPR 2023
> >
> > [8]  Xiaotong Li, Yongxing Dai, Yixiao Ge, Jun Liu, Ying Shan, and Ling-Yu Duan. Uncertainty modeling for out-of-distribution generalization. ICLR, 2022.

---

> > > ### Comment · Reviewer_EPCk · 2024-08-10
> > > **Resonse to the comment**
> > >
> > > Dear authors,
> > > Thanks for the comment. I don't have further questions. Considering the supplemented results, I will raise my score.

---

### Author Rebuttal · Authors · 2024-08-06

We would like to thank the ACs and reviewers for their constructive comments on our paper. We are encouraged by the positive feedback, including remarks such as "the aspect is important", "this paper contains novelty and is technically sound", "the theoretical analysis is a significant contribution to the field", and "the method is novel and mathematically grounded".

In our rebuttal, we answer the reviewers' concerns point-by-point, covering: 1) explanations of the theoretical analysis, 2) differences and comparisons with related works such as DGMamba, 3) protection of semantic features in input-dependent matrices, 4) additional ablation studies, 5) implementation details, and 6) scalability of our analysis and method.

Due to the $6,000$-character limit for responses to each reviewer, we have included the "global_tables.pdf" file, which contains all relevant tables. We sincerely hope our responses could address all concerns and look forward to further discussion during the author-reviewer discussions phase.

---

### Decision · Program_Chairs · 2024-09-25

**Decision:**

Accept (poster)

**Comment:**

The paper receives all positive ratings after rebuttal, where three reviewers upgraded the rating. Overall, the authors have provided more technical explanations (e.g., theoretical clarification, comparisons with the DGMamba method) and experimental results (e.g., computational cost, results with ViT, comparisons with other feature identification methods) in the rebuttal. The AC carefully read the paper, reviews, and replies, and agrees with the reviewers' feedback. Hence the AC recommends the acceptance decision, based on the merit where the paper designs a Mamba framework for improving domain generalization with theoretical analysis and extensive experimental results to verify the proposed method. Also, it would be highly recommended that the authors include suggested feedback in the final version and release the code to the public for fostering the progress in this field.